# ZENO++: ROBUST FULLY ASYNCHRONOUS SGD

## ABSTRACT

We propose Zeno++, a new robust asynchronous Stochastic Gradient Descent (SGD) procedure which tolerates Byzantine failures of the workers. In contrast to previous work, Zeno++ removes some unrealistic restrictions on worker-server communications, allowing for fully asynchronous updates from anonymous workers, arbitrarily stale worker updates, and the possibility of an unbounded number of Byzantine workers. The key idea is to estimate the descent of the loss value after the candidate gradient is applied, where large descent values indicate that the update results in optimization progress. We prove the convergence of Zeno++ for non-convex problems under Byzantine failures. Experimental results show that Zeno++ outperforms existing approaches.

## 1 INTRODUCTION

Synchronous training and asynchronous training are the two most common paradigms of distributed machine learning. On the one hand, synchronous training requires, periodically, the global updates at the server to be blocked until all the workers respond. In contrast, for asynchronous training, the server updates the global model immediately after a worker responds. Theoretical and experimental analysis (Dutta et al., 2018) suggests that synchronous training is more stable with less noise, but can also be slowed down by the global barrier across all the workers. Asynchronous training is generally faster, but needs to address instability and noisiness due to staleness. In this paper, we focus on asynchronous training.

We study the security of distributed asynchronous Stochastic Gradient Descent (SGD) in a centralized worker-server architecture, also known as the Parameter Server (or PS) architecture. In the PS architecture, there are server nodes and worker nodes. When combined with an asynchronous SGD approach, each worker pulls the global model from the servers, estimates the gradients using the local portion of the training data, then sends the gradient estimates to the servers. The servers update the model as soon as a new gradient is received from any worker.

The security of machine learning has gained increasing attention in recent years. In particular, tolerance to Byzantine failures (Blanchard et al., 2017; Chen et al., 2017; Yin et al., 2018; Feng et al., 2014; Su and Vaidya, 2016a;b; Xie et al., 2018b; Alistarh et al., 2018; Cao and Lai, 2018) has become an important topic in the distributed machine learning literature. Byzantine failures are well-studied for the distributed systems (Lamport et al., 1982a). However, in distributed machine learning, Byzantine failures have unique properties. In brief, the goal of Byzantine workers is to prevent convergence of the model training. By construction, Byzantine failures (Lamport et al., 1982b) assume the worst case, i.e., the Byzantine workers can behave arbitrarily. Such failures may be caused by a variety of reasons including but not limited to: hardware/software bugs, vulnerable communication channels, poisoned datasets, or malicious attackers. To make things worse, groups of Byzantine workers can collude, potentially resulting in more harmful attacks. It is also clear that as the worst case, Byzantine failures generalize benign failures such as hardware or software errors.

Unlike previous work (Damaskinos et al., 2018), we tackle Byzantine tolerance in a more general scenario. The Byzantine tolerance of asynchronous SGD is challenging in this case because of:

- **Asynchrony.** The lack of synchrony incurs additional noise for the stochastic gradients. Such noise makes it more difficult to distinguish the Byzantine gradients from the benign ones, especially as Byzantine behavior may exacerbate staleness.

- **Unpredictable successive updates.** The lack of synchronous scheduling makes it possible for the server to receive updates from Byzantine workers successively. Thus, even in the standard scenario,

where less than half of the workers are Byzantine, the server can be suffocated by successive Byzantine gradients.

- **Unbounded number of Byzantine workers.** For the Byzantine tolerance in fully asynchronous training, the assumption of a bounded number of Byzantine workers is meaningless. In Byzantine-tolerant synchronous training (Blanchard et al., 2017; Chen et al., 2017; Yin et al., 2018; Xie et al., 2018b;a), the servers can compare the candidate gradients with each other, and utilize the majority assumption to filter out the harmful gradients, or use robust aggregation to bound the error. However, such strategies are infeasible in asynchronous training, since there is nothing to compare to or aggregate. Aggregating the successive gradients is also meaningless since the successive gradients could all be pushed by the same Byzantine worker. Furthermore, although most of the previous work (Blanchard et al., 2017; Chen et al., 2017; Yin et al., 2018; Feng et al., 2014; Su and Vaidya, 2016a;b; Alistarh et al., 2018; Cao and Lai, 2018) assumes a majority of honest workers, this requirement is not guaranteed to be satisfied in practice.

The key idea of our approach is to estimate the descent of the loss value after the candidate gradient is applied to the model parameters, based on the Byzantine-tolerant synchronous SGD algorithm, Zeno (Xie et al., 2018b). Intuitively, if the loss value decreases, the candidate gradient is likely to result in optimization progress. For computational efficiency, we also propose a lazy update.

To the best of our knowledge, this paper is the first to theoretically *and* empirically study Byzantine-tolerant fully asynchronous SGD with anonymous workers, and potentially an unbounded number of Byzantine workers. In summary, our contributions are:

- We propose Zeno++, a new approach for Byzantine-tolerant fully asynchronous SGD with anonymous workers.
- We show that Zeno++ tolerates Byzantine workers without any limit on either the staleness or the number of Byzantine workers.
- We prove the convergence of Zeno++ for non-convex problems.
- Experimental results validate that 1) existing algorithms may fail in practical scenarios, and 2) Zeno++ gracefully handles such cases.

## 2 RELATED WORK

Most of the existing Byzantine-tolerant SGD algorithms focus on synchronous training. Chen et al. (2017); Su and Vaidya (2016a;b); Yin et al. (2018); Xie et al. (2018a) use robust statistics (Huber, 2011) including the geometric median, coordinate-wise median, and trimmed mean as Byzantine-tolerant aggregation rules. Blanchard et al. (2017); Mhamdi et al. (2018) propose Krum and its variants, which select the candidates with minimal local sum of Euclidean distances. Alistarh et al. (2018) utilize historical information to identify harmful gradients. Chen et al. (2018) use coding theory and majority voting to recover correct gradients. Most of these synchronous algorithms assume that most of the workers are non-Byzantine. However, in practice, there are no guarantees that the number of Byzantine workers can be controlled. Xie et al. (2018b); Cao and Lai (2018) propose synchronous SGD algorithms for an unbounded number of Byzantine workers.

Recent years have witnessed an increasing number of large-scale machine learning algorithms, including asynchronous SGD (Zinkevich et al., 2009; Lian et al., 2018; Zheng et al., 2017; Zhou et al., 2018). Damaskinos et al. (2018) proposed Kardam, which to our knowledge is the only prior work to address Byzantine-tolerant asynchronous training. Kardam utilizes the Lipschitzness of the gradients to filter out outliers. However, Kardam assumes a threat model much weaker than ours. The major differences in the threat model are listed as follows:

- **Verification of worker identity.** Unlike Kardam, we do not require verifying the identities of the workers when the server receives gradients. Kardam uses the so-called *empirical Lipschitz coefficient*, to test the benignity of the gradient sent by a specific worker. Such a mechanism keeps the record of the *empirical Lipschitz coefficient* of each worker. Thus, whenever a gradient is received, the Kardam server must be able to identify the identity/index of the worker. However, since Byzantine workers can behave arbitrarily, they can fake their identities/indices when sending gradients to the servers. Thus, Kardam assumes a threat model much weaker than the traditional

Byzantine failure/threat model. Note that for synchronous training, the server can partially counter the index spoofing attack by simply filtering out all the gradients with duplicated indices. However, such an approach is infeasible for asynchronous training.

- **Bounded staleness of workers/limit of successive gradients.** Unlike Kardam, we do not require bounded staleness of the workers. Kardam requires that the number of gradients successively received from a single worker is bounded above. To be more specific, on the server, any sequence of successively received gradients of length $2q + 1$ must contain at least $q + 1$ gradients from honest workers. However, in real-world asynchronous training, such an assumption is very difficult to satisfy.

- **A majority of honest workers.** Unlike Kardam, we do not require a majority of honest workers. Kardam requires that the number of Byzantine workers is less than one-third of the total number of workers – much stronger restriction than the standard setting that allows for the number of Byzantine workers to be up to 50% of the total number of workers. Zeno++ further extends this guarantee to allow for not only 50%, but also a majority of Byzantine workers.

## 3 MODEL

We consider the following optimization problem: $\min_{x \in \mathbb{R}^d} F(x)$, where $F(x) = \frac{1}{m} \sum_{i \in [m]} \mathbb{E}_{z_i \sim \mathcal{D}_i} f(x; z_i)$, for $\forall i \in [m]$, $z_i$ is sampled from the local data $\mathcal{D}_i$ on the $i$th device.

We solve this problem in a distributed manner with $m$ workers. Each worker trains the model on local data. In each iteration, the $i$th worker will sample $n$ independent data points from the dataset $\mathcal{D}_i$, and compute the gradient of the local empirical loss $F_i(x) = \frac{1}{n} \sum_{j=1}^{n} f(x; z_{i,j}), \forall i \in [m]$, where $z_{i,j} \sim \mathcal{D}_i$ is the $j$th sampled data on the $i$th worker. When there are no Byzantine failures, the servers update the model whenever a new gradient is received:

$$x_{t+1} = x_t - \gamma_t g_\tau, \quad g_\tau = \frac{1}{n} \sum_{j \in [n]} \nabla f(x_\tau; z_{i,j}), \tau \le t, i \in [m].$$

When there are Byzantine failures, $g_\tau$ can be replaced by arbitrary value (Damaskinos et al., 2018). Formally, we define the threat model as follows.

**Definition 1.** *(Threat Model). When the server receives a gradient estimator $\tilde{g}_\tau$, it is either correct or Byzantine. If sent by a Byzantine worker, $\tilde{g}_\tau$ is assigned arbitrary value. If sent by an honest worker, the correct gradient is $\frac{1}{n} \sum_{j=1}^{n} \nabla f(x_\tau; z_{i,j}), \tau \le t, i \in [m]$. Thus, we have*

$$\tilde{g}_\tau = \begin{cases} arbitrary\ value, & if\ sent\ by\ a\ Byzantine\ worker, \\ \frac{1}{n} \sum_{j=1}^{n} \nabla f(x_\tau; z_{i,j}), \tau \le t, i \in [m], & otherwise. \end{cases}$$

*We assume that $q$ out of $m$ workers are Byzantine, where $q \le m$. Furthermore, the indices of Byzantine workers can change across different iterations.*

Table 1: Notation

| Notation | Description |
|---|---|
| $m, [m], q$ | Number of workers, set of integers $\{1, \ldots, m\}$, number of Byzantine workers |
| $\mathcal{D}_i, \mathcal{S}$ | $\mathcal{D}_i$ is the training dataset on the $i$th worker, $\mathcal{S}$ is the validation dataset on *Zeno++* server |
| $n, n_s$ | Mini-batch size of workers, mini-batch size of `Zeno++` server |
| $T, t, \gamma$ | Number of global iterations, index of global iteration, learning rate |
| $\rho, \epsilon, k$ | Hyperparameter of Zeno++, $k$ is the maximum delay of $g_r$, also called "server delay" |
| $k_w$ | Maximum delay of workers, also called "worker delay", different from the "server delay" $k$ |
| $\|\cdot\|$ | All the norms in this paper are $l_2$-norms |

## 4 METHODOLOGY

In this section, we introduce `Zeno++`, a Byzantine-tolerant asynchronous SGD algorithm based on inner-product validation. `Zeno++` is a computationally efficient version of its prototype: `Zeno+`.

### 4.1 ZENO+

Like `Zeno` (Xie et al., 2018b), we compute a score for each candidate gradient estimator by using the stochastic zero-order oracle. However, in contrast to the existing synchronous SGD with majority-

based aggregation methods, we need a hard threshold to decide whether a gradient is accepted, as sorting is not meaningful in asynchronous settings. This descent score is described next.

**Definition 2.** *(Stochastic Descent Score (Xie et al., 2018b)) Denote* $f_s(x) = \frac{1}{n_s} \sum_{j=1}^{n_s} f(x; z_j)$, *where $z_j$'s are i.i.d. samples drawn from $\mathcal{S}$, where $\mathcal{S} \neq \mathcal{D}_i, \forall i \in [m]$, and $n_s$ is the batch size of $f_s(\cdot)$. For any gradient estimator (correct or Byzantine) g, model parameter x, learning rate $\gamma$, and a constant weight $\rho > 0$, we define its stochastic descent score as follows:*

$$Score_{\gamma,\rho}(g, x) = f_s(x) - f_s(x - \gamma g) - \rho \|g\|^2.$$

**Remark 1.** *Note that we assume that the dataset $\mathcal{S}$ for computing $f_s(\cdot)$ is different from the training dataset, e.g., can be a separated validation dataset. In other words, $\mathcal{S} \neq \mathcal{D}_1 \neq \cdots \neq \mathcal{D}_m \neq \cup_{i=1}^m \mathcal{D}_i$.*

The score defined in Definition 2 is composed of two parts: the estimated descent of the loss function, and the magnitude of the update. The score increases when the estimated descent of the loss function, $f_s(x) - f_s(x - \gamma g)$, gets larger. We penalize the score by $-\rho \|g\|^2$, so that the change of the model parameter will not be too large. A large descent suggests faster convergence. Observe that even when a gradient is Byzantine, a small magnitude indicates that it will be less harmful to the model.

Using the *stochastic descent score*, we can set a hard threshold parameterized by $\epsilon$ to filter out candidate gradients with relatively small scores. The detailed algorithm is outlined in Algorithm 1.

---

**Algorithm 1** Zeno+

**Server:**
1: $x_0 \leftarrow rand(), t \leftarrow 1$  $\triangleright$ Initialization
2: **repeat**
3:    Randomly sample $z_j \sim \mathcal{S}, \forall j \in [n_s]$ to compute $f_s$ (Note: $\mathcal{S} \neq \mathcal{D}_1 \neq \cdots \neq \mathcal{D}_m$)
4:    Receive $\tilde{g}$ from an arbitrary worker
5:    Normalize $g = c\tilde{g}$ such that $\|g\|^2 = \|\nabla f_s(x_{t-1})\|^2, c > 0$
6:    **if** $Score_{\gamma,\rho}(g, x_{t-1}) \geq -\gamma\epsilon$ **then**
7:       $x_t \leftarrow x_{t-1} - \gamma g, t \leftarrow t + 1$
8:    **end if**
9: **until** Convergence

**Worker** $i = 1, \ldots, m$**:**
1: **function** WORKER (HONEST)
2:    **repeat**
3:       Pull $x_\tau$ from the server
4:       Draw random samples $z_{i,j} \sim \mathcal{D}_i, \forall j \in [n]$, compute $\tilde{g} \leftarrow \frac{1}{n} \sum_{j \in [n]} \nabla f(x_\tau; z_{i,j})$
5:       Push $\tilde{g}$ to the server
6:    **until** Convergence
7: **end function**

---

### 4.2   ZENO++

Calculating the *stochastic descent score* for every candidate gradient can be computationally expensive. To reduce the computation overhead, we approximate it by its first-order Taylor's expansion.

**Definition 3.** *(Approximated Stochastic Descent Score) Denote* $f_s(x) = \frac{1}{n_s} \sum_{j=1}^{n_s} f(x; z_j)$, *where $z_j$'s are i.i.d. samples drawn from $\mathcal{S}$, where $\mathcal{S} \neq \mathcal{D}_i, \forall i \in [m]$, and $n_s$ is the batch size of $f_s(\cdot)$. For any gradient estimator (correct or Byzantine) g, model parameter x, learning rate $\gamma$, and a constant weight $\rho > 0$, we approximate its stochastic descent score as follows:*

$$Score_{\gamma,\rho}(g, x) \approx \gamma \langle \nabla f_s(x), g \rangle - \rho \|g\|^2.$$

In brief, `Zeno++` is a computationally efficient version of `Zeno+` which uses this *approximated stochastic descent score*, combined with lazy updates. The detailed algorithm is shown in Algorithm 2. Compared to `Zeno`, we highlight several new techniques in `Zeno++` (Algorithm 2), specially designed for asynchronous training: 1) re-scaling the candidate gradient (Line 6); 2) first-order Taylor's expansion (Line 7); 3) hard threshold instead of comparison with the others (Line 7); 4) lazy update for reducing the computation overhead (Line 9).

Before moving forward, we wish to highlight several practical remarks for `Zeno++`:

- **Preparing the validation dataset for Zeno++:** The dataset $\mathcal{S}$ used for calculating $v$ (the validation gradient of `Zeno++`) can be collected in many ways. It can be a separate validation dataset provided by a trusted third party. Another reasonable choice is that, a group of trusted workers can upload local data perturbed by additional noise (to help protect the users' privacy). Typically, the validation dataset is small and different from the training dataset, thus can only be used to validate the gradients, and cannot be directly used for training, as shown in Section 6.

- **Scheduling** $ZenoUpdater(x)$**:** $ZenoUpdater(x)$ updates $v$ in the background. It will only be triggered when the global model $x_t$ is updated and the server is idle. Another scheduling strategy is to trigger $ZenoUpdater(x)$ after every $k$ iterations. Thus, $k$ is the upper bound of the delay of $v$. A reasonable choice is $k = m$, so that ideally $v$ is updated after all the $m$ workers respond.

- **Computational efficiency:** We can reduce the computation overhead of the `Zeno++` server by decreasing the mini-batch size $n_s$, or the frequency of the activation of $ZenoUpdater(x)$. However, doing so will potentially incur larger noise for $v$, which makes a trade-off.

---

**Algorithm 2** Zeno++

**Server:**

1: $x_0 \leftarrow rand(), t \leftarrow 1$                                         ▷ Initialization
2: **repeat**
3:     **repeat**
4:         Receive $\tilde{g}$ from an arbitrary worker
5:         Read $v$ with lock ($v$ may be from an old version of $x$: $v = \nabla f_s(x_\tau), \tau \leq t - 1$)
6:         Normalize $g = c\tilde{g}$ such that $\|g\|^2 = \|v\|^2, c > 0$
7:     **until** $\gamma \langle v, g \rangle - \rho\|g\|^2 \geq -\gamma\epsilon$
8:     $x_t \leftarrow x_{t-1} - \gamma g, \quad t \leftarrow t + 1$
9:     Lazy update of $v$: Run non-blocking $ZenoUpdater(x_t)$, if idle, or after every $k$ iterations
10: **until** Convergence
11: **function** ZENOUPDATER$(x)$
12:     Randomly sample $z_j \sim \mathcal{S}, \forall j \in [n_s]$ to compute $f_s$ (Note: $\mathcal{S} \neq \mathcal{D}_1 \neq \cdots \neq \mathcal{D}_m$)
13:     Write with lock: $v \leftarrow \nabla f_s(x) = \frac{1}{n_s}\sum_{j=1}^{n_s} \nabla f(x; z_j)$
14: **end function**

**Worker** $i = 1, \ldots, m$**:**

1: **function** WORKER (HONEST)
2:     **repeat**
3:         Pull $x_\tau$ from the server
4:         Draw random samples $z_{i,j} \sim \mathcal{D}_i, \forall j \in [n]$, compute $\tilde{g} \leftarrow \frac{1}{n}\sum_{j \in [n]} \nabla f(x_\tau; z_{i,j})$
5:         Push $\tilde{g}$ to the server
6:     **until** Convergence
7: **end function**

---

## 5 THEORETICAL GUARANTEES

In this section, we prove the convergence of `Zeno++` (Algorithm 2) under Byzantine failures. We start with definitions used in the convergence analysis.

**Definition 4.** *(Smoothness) Differentiable $f(x)$ satisfies L-smoothness if there exists $L > 0$ such that $\forall x, y, f(y) - f(x) \leq \langle \nabla f(x; z), y - x \rangle + \frac{L}{2}\|y - x\|^2$.*

**Definition 5.** *(Polyak-Łojasiewicz (PL) inequality) Differentiable $f(x)$ satisfies the PL inequality (Polyak, 1963) if there exists $\mu > 0$, such that $\forall x: f(x) - f(x_*) \leq \frac{1}{2\mu}\|\nabla f(x)\|^2$.*

### 5.1 CONVERGENCE GUARANTEES

We prove the convergence of Algorithm 2 for non-convex problems with the following assumption.

**Assumption 1.** *(Bounded server delay) For `Zeno++`, we assume that the delay of the validation gradient $v$ is upper-bounded. Without loss of generality, suppose the current model is $x_t$, and $v = \nabla f_s(x_\tau)$, where $\tau \leq t$. We assume that for $\forall t, t - \tau \leq k$.*

**Remark 2.** *Zeno++ does not require bounded delay for the workers. The bounded delay requirement in Assumption 1 is only for the validation gradient $v$ on the server, not for the workers.*

We first analyze the convergence of functions that satisfy the PL inequality.

**Theorem 1.** *Assume that $F(x)$ and $f_s(x)$ are L-smooth and satisfy the PL inequality. Assume that for $\forall x$, the correct gradients and validation gradients are upper-bounded: $\|\nabla F(x)\|^2 \leq V_1$, $\|\nabla f_s(x)\|^2 \leq V_1$, and the validation gradients are always non-zero and lower-bounded: $\|\nabla f_s(x)\|^2 \geq V_2$, where $0 < V_2 \leq V_1$. Furthermore, we assume that the validation set is close to the training set, which implies bounded variance: $\mathbb{E}\left[\|\nabla f_s(x) - \nabla F(x)\|^2\right] \leq V_3, \forall x$. Taking $\gamma < \min(1, \frac{1}{L})$ and $\rho \geq \frac{\alpha\sqrt{\gamma}V_1}{2\mu V_2}$, after $T$ global updates, Algorithm 2 converges to a global optimum:*

$$\mathbb{E}\left[F(x_T) - F(x_*)\right] \leq (1 - \alpha\sqrt{\gamma})^T \left[F(x_0) - F(x_*)\right] + \frac{\sqrt{\gamma}}{\alpha}\mathcal{O}(k^2 V_1 + V_3 + \epsilon).$$

**Remark 3.** *The assumption of the lower bounded gradient $\|\nabla f_s(x)\|^2 \geq V_2$ is necessary. We need $\nabla f_s(x) \neq 0$, so that the normalization in Line 6 and inner product in Line 7 of Algorithm 2 are feasible. In practice, if we have a mini-batch with zero gradient $\nabla f_s(x) = 0$ on server, we can simply draw additional samples and add them to the mini-batch, until such gradient is non-zero.*

For general smooth but non-convex functions, we have the following convergence guarantee.

**Theorem 2.** *Assume that $F(x)$ and $f_s(x)$ are L-smooth and potentially non-convex. Assume that for $\forall x$, the true gradients and validation gradients are upper-bounded: $\|\nabla F(x)\|^2 \leq V_1$, $\|\nabla f_s(x)\|^2 \leq V_1$, and the validation gradients are always non-zero and lower-bounded: $\|\nabla f_s(x)\|^2 \geq V_2$, where $0 < V_2 \leq V_1$. Furthermore, we assume that the validation set is close to the training set, which implies bounded variance: $\mathbb{E}\left[\|\nabla f_s(x) - \nabla F(x)\|^2\right] \leq V_3, \forall x$. Taking $\gamma < \min(1, \frac{1}{L})$ and $\rho \geq \frac{\alpha\sqrt{\gamma}V_1}{V_2}$, after $T$ global updates, Algorithm 2 converges to a critical point:*

$$\frac{\mathbb{E}\left[\sum_{t \in [T]} \|\nabla F(x_{t-1})\|^2\right]}{T} \leq \frac{\mathbb{E}\left[F(x_0) - F(x_*)\right]}{\alpha\sqrt{\gamma}T} + \frac{\sqrt{\gamma}}{\alpha}\mathcal{O}(k^2 V_1 + V_3 + \epsilon).$$

*Furthermore, if we take $\gamma = \frac{1}{LT}$, then we have $\frac{\mathbb{E}\left[\sum_{t \in [T]} \|\nabla F(x_{t-1})\|^2\right]}{T} \leq \mathcal{O}\left(\frac{1}{\alpha\sqrt{T}}\right)$.*

**Remark 4.** *$\rho$ controls the trade-off between the acceptance ratio and the convergence rate. Large positive $\rho$ makes the convergence faster, but fewer candidate gradients pass the test of Zeno++. Small positive $\rho$ increases the acceptance ratio, but may also potentially slow down the convergence or incur larger variance. We use $\alpha > 0$ to bridge $\rho$ to the convergence rate and the variance. Larger $\alpha$ makes $\rho$ larger, which improves the convergence rate, but also enlarges the variance. Using non-zero $\epsilon$ potentially results in negative thresholds, which enlarges the acceptance ratio, but also increases the false negative ratio (the ratio of Byzantine gradients that are not filtered out by Zeno++).*

## 6 EXPERIMENTS

In this section, we evaluate the proposed algorithm, `Zeno++`. Note that we do not evaluate the prototype algorithm `Zeno+`, since its computation overhead is too large for practical settings. Due to the space limitation, zoomed figures and additional experiments (including evaluation on an additional label-flipping attack, and testing the sensitivity to hyperparameters) are presented in the appendix.

### 6.1 DATASETS AND EVALUATION METRICS

We conduct experiments on the benchmark CIFAR-10 image classification dataset (Krizhevsky and Hinton, 2009), which is composed of 50k images for training and 10k images for testing. We use a convolutional neural network (CNN) with 4 convolutional layers followed by 2 fully connected layers. The detailed network architecture can be found in our submitted source code (will be released upon publication). In the 50k images for training, we randomly extracted 2.5k of them as the validation set for `Zeno++`, the remaining are randomly partitioned onto all the workers. In each experiment, we launch 10 worker processes. We repeat each experiment 10 times and take the average. Each experiment is composed of 200 epochs, where each epoch is a full pass of the training dataset. We simulate asynchrony by drawing random delay from a uniform distribution in the range of $[0, k_w]$, where $k_w$ is the maximum worker delay (different from the maximum server delay $k$ of `Zeno++`).

We use top-1 accuracy on the testing set and the cross-entropy loss function on the training set as the evaluation metrics. We also report the false positive rate (FP), which is the ratio of correct gradients that are recognized as Byzantine and filtered out by `Zeno++` or the `Kardam` baseline.

### 6.1.1 BASELINES

We use the asynchronous SGD without failures/attacks as the gold standard, which is referred to as `AsyncSGD without attack`. Since `Kardam` is the only previous work on Byzantine-tolerant asynchronous SGD, we use it as the baseline.

One may conjecture that `Zeno++` is analogous to training on the validation data. To explore this, we consider training only on $\mathcal{S}$ – assumed to be clean data on the server, i.e., update the model only using $v = \nabla f_s(x)$ on the server, without using any workers. We call this baseline `Server-only`. We fine-tune the learning rate and show the best results of `Server-only`.

### 6.2 NO ATTACK

We first test the convergence when there are no attacks. In all the experiments, we take the learning rate $\gamma = 0.1$, mini-batch size $n = n_s = 128$, $\rho = 0.002$, $\epsilon = 0.1$, $k = 10$. For `Kardam`, we take $q = 2$ (i.e. here `Kardam` assumes that there are 2 Byzantine workers). The result is shown in Figure 1. `Zeno++` converges a little bit slower than `AsyncSGD`, but faster than `Kardam`, especially when the worker delay is large. When $k_w = 10$, `Zeno++` converges much faster than Kardam. `Server-only` performs badly on both training and testing data.

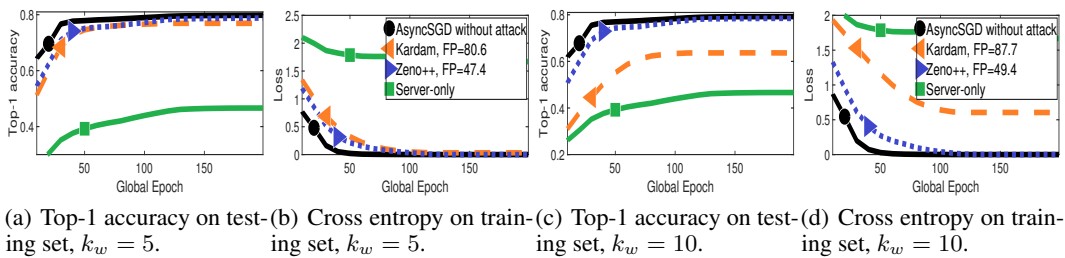

(a) Top-1 accuracy on test-ing set, $k_w = 5$.    (b) Cross entropy on train-ing set, $k_w = 5$.    (c) Top-1 accuracy on test-ing set, $k_w = 10$.    (d) Cross entropy on train-ing set, $k_w = 10$.

Figure 1: Convergence without attacks, with different maximum worker delays $k_w$. $\rho = 0.002, \epsilon = 0.1, k = 10$ for `Zeno++`. *FP* refers to the fraction of false positive detect ions i.e. incorrect prediction that a message is Byzantine.

### 6.3 SIGN-FLIPPING ATTACK

We test the Byzantine-tolerance to the "sign-flipping" attack, which was proposed in Damaskinos et al. (2018). In such attacks, the Byzantine workers send $-10\nabla f(x)$ instead of the correct gradient $\nabla f(x)$ to the server. In all the experiments, we take the learning rate $\gamma = 0.1$, mini-batch size $n = n_s = 128$, $\rho = 0.002$, $\epsilon = 0.1$, $k = 10$. The result is shown in Figure 2, with different number of Byzantine workers $q$. It is shown that when $q = 4$, `Zeno++` converges slightly slower than `AsyncSGD` without attacks, and much faster than `Kardam`. Actually, we observe that `Kardam` fails to make progress when the worker delay is large. When the number of Byzantine workers gets larger ($q = 8$), the convergence of `Zeno++` gets slower, but it still makes reasonable progress, while `AsyncSGD` and `Kardam` fail. Note that `Kardam` performs even worse than `Server-only`, which means that `Kardam` is not even as good as training on a single honest worker. Thus, when there are Byzantine workers, distributed training with `Kardam` is meaningless.

### 6.4 DISCUSSION

`Kardam` performs surprisingly badly in our experiments. The experiments in Damaskinos et al. (2018) focus on dampening staleness when there are no Byzantine failures. For Byzantine tolerance, Damaskinos et al. (2018) only reports that `Kardam` filters out 100% of the Byzantine gradients, which matches the results in our experiments. However, we observe that in addition to filtering out 100% of the Byzantine gradients, `Kardam` also filters nearly 100% of the correct gradients. In Figure 2, we

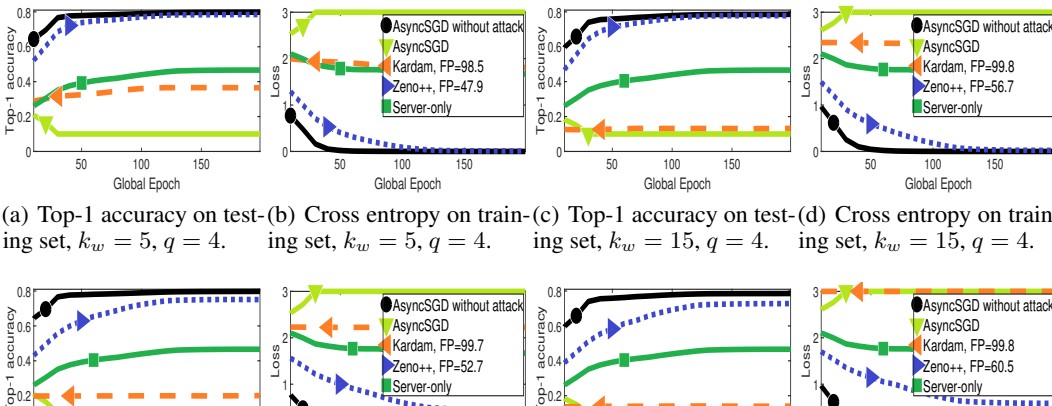

(a) Top-1 accuracy on test-ing set, $k_w = 5, q = 4$.

(b) Cross entropy on train-ing set, $k_w = 5, q = 4$.

(c) Top-1 accuracy on test-ing set, $k_w = 15, q = 4$.

(d) Cross entropy on train-ing set, $k_w = 15, q = 4$.

(e) Top-1 accuracy on test-ing set, $k_w = 5, q = 8$.

(f) Cross entropy on train-ing set, $k_w = 5, q = 8$.

(g) Top-1 accuracy on test-ing set, $k_w = 15, q = 8$.

(h) Cross entropy on train-ing set, $k_w = 15, q = 8$.

Figure 2: Convergence with sign-flipping attacks with different maximum worker delays $k_w$. For any correct gradient $g$, if selected to be Byzantine, $g$ will be replaced by $-10g$. $q \in \{4, 8\}$ out of the 10 workers are Byzantine. $\rho = 0.002, \epsilon = 0.1, k = 10$ for Zeno++. *FP* refers to the fraction of false positive detect ions i.e. incorrect prediction that a message is Byzantine.

report that the false positive rate of Kardam is nearly 99%, which makes the convergence extremely slow. To make things worse, Kardam does not even perform as good as Server-only, which makes the distributed training with Kardam totally meaningless. One reason why Kardam performs badly is that we use a more general threat model in this paper, which does not guarantee an important assumption of Kardam, namely "any sequence of successively received gradients of length $2q + 1$ must contain at least $q + 1$ gradients from honest workers". It is clear that this assumption is quite strong, as in an asynchronous setting, Byzantine workers can easily send long sequences of erroneous responses. Our approach does not depend on such a strong assumption.

In all the experiments, Zeno++ converges faster than the baselines when there are Byzantine failures. Although the convergence of Zeno++ is slower than AsyncSGD when there are no attacks, we find that it provides a reasonable trade-off between security and convergence speed. In general, larger worker delay $k_w$ and more Byzantine workers $q$ add more error and noise to the gradients, which slows down the convergence, because there are fewer valid gradients for the server to use. Zeno++ can filter out most of the harmful gradients at the cost of $FP \approx 50\%$.

Note that Server-only is an extreme case that only uses the server and the validation dataset to train the model in a non-distributed manner, which will not be affected by Byzantine workers. However, only using the validation data is not enough for training, as shown in Figure 1. Similarly in practice, we can use a small dataset separated from the training data for cross-validation, but will never directly train the model only on such validation dataset. Furthermore, as shown in Figure 2, Zeno++ performs much better than Server-only. Thus, we can draw to the conclusion that Zeno++ is efficiently training the model on the honest workers in a distributed manner, which is not equivalent to training on the validation dataset only.

On average, the server computes $\frac{n_s}{k} = 12.8$ gradients in each iteration, since the validation gradient $v$ of Zeno++ is updated after every $k = 10$ iterations. Thus, the workload on the server is much smaller than a worker. Furthermore, since we can parallelize the workload on the server and workers, the computation overhead of $v$ can be hidden, so that Zeno++ can benefit from distributed training.

## 7 CONCLUSION

We propose a novel Byzantine-tolerant fully asynchronous SGD algorithm: Zeno++. The algorithm provably converges. Our empirical results show good performance compared to previous work. In future work, we will explore variations of our approach for other settings such as federated learning.

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

# Appendix

## A    PROOFS

### A.1    ZENO++

We first analyze the convergence of the functions whose gradients grow as a quadratic function of sub-optimality.

**Theorem 1.** *Assume that $F(x)$ and $f_s(x)$ have L-smoothness and PL inequality (potentially non-convex). Assume that for $\forall x$, the true gradients and stochastic gradients are upper-bounded: $\|\nabla F(x)\|^2 \leq V_1$, $\|\nabla f_s(x)\|^2 \leq V_1$, and the stochastic gradients for Zeno testing are always non-zero and lower-bounded: $\|\nabla f_s(x)\|^2 \geq V_2$, where $0 < V_2 \leq V_1$. Furthermore, we assume that the validation set is close to the training set, which implies bounded variance: $\mathbb{E}\left[\|\nabla f_s(x) - \nabla F(x)\|^2\right] \leq V_3, \forall x$. Taking $\gamma < \min(1, \frac{1}{L})$ and $\rho \geq \frac{\alpha\sqrt{\gamma}V_1}{2\mu V_2}$, after $T$ global updates, Algorithm 2 converges to a global optimum:*

$$\mathbb{E}\left[F(x_T) - F(x_*)\right] \leq (1 - \alpha\sqrt{\gamma})^T \left[F(x_0) - F(x_*)\right] + \frac{\sqrt{\gamma}}{\alpha}\mathcal{O}(k^2 V_1 + V_3 + \epsilon).$$

*Proof.* If any gradient estimator $g$ passes the test of `Zeno++`, then we have
$$\langle \nabla f_s(x_\tau), -\gamma g \rangle \leq -\rho\|g\|^2 + \gamma\epsilon,$$
where $\tau \leq t - 1$.

Thus, we have
$$\langle \nabla F(x_{t-1}), -\gamma g \rangle$$
$$\leq \langle \nabla F(x_{t-1}) - \nabla f_s(x_\tau), -\gamma g \rangle - \rho\|\nabla f_s(x_\tau)\|^2 + \gamma\epsilon$$
$$\leq -\rho\frac{V_2}{V_1}\|\nabla F(x_{t-1})\|^2 + \frac{\gamma}{2}\|\nabla F(x_{t-1}) - \nabla f_s(x_\tau)\|^2 + \frac{\gamma}{2}\|\nabla f_s(x_\tau)\|^2 + \gamma\epsilon$$
$$\leq -\rho\frac{V_2}{V_1}\|\nabla F(x_{t-1})\|^2 + \gamma\|\nabla F(x_\tau) - \nabla f_s(x_\tau)\|^2 + \gamma\|\nabla F(x_{t-1}) - \nabla F(x_\tau)\|^2 + \frac{\gamma}{2}V_1 + \gamma\epsilon$$
$$\leq -\rho\frac{V_2}{V_1}\|\nabla F(x_{t-1})\|^2 + \gamma V_3 + \gamma\|\nabla F(x_{t-1}) - \nabla F(x_\tau)\|^2 + \frac{\gamma}{2}V_1 + \gamma\epsilon.$$

$\|\nabla F(x_{t-1}) - \nabla F(x_\tau)\|^2$ can be upper-bounded using $L$-smoothness and the bounded delay:
$$\|\nabla F(x_{t-1}) - \nabla F(x_\tau)\|^2 \leq L^2\|x_{t-1} - x_\tau\|^2 \leq L^2 k^2 \gamma^2 V_1.$$

Again, using smoothness, taking $\rho \geq \frac{\alpha\sqrt{\gamma}V_1}{2\mu V_2}$, we have
$$\mathbb{E}\left[F(x_t) - F(x_*)\right]$$
$$\leq F(x_{t-1}) - F(x_*) + \langle \nabla F(x_{t-1}), -\gamma\mathbb{E}[g] \rangle + \frac{L\gamma^2}{2}\mathbb{E}\|g\|^2$$
$$\leq F(x_{t-1}) - F(x_*) - \rho\frac{V_2}{V_1}\|\nabla F(x_{t-1})\|^2 + \gamma V_3 + L^2 k^2 \gamma^3 V_1 + \frac{\gamma}{2}V_1 + \frac{L\gamma}{2}V_1 + \gamma\epsilon$$
$$\leq F(x_{t-1}) - F(x_*) - \rho\frac{V_2}{V_1}\|\nabla F(x_{t-1})\|^2 + \gamma\mathcal{O}(k^2 V_1 + V_3 + \epsilon)$$
$$\leq \left[1 - 2\mu\rho\frac{V_2}{V_1}\right]\left[F(x_{t-1}) - F(x_*)\right] + \gamma\mathcal{O}(k^2 V_1 + V_3 + \epsilon)$$
$$\leq (1 - \alpha\sqrt{\gamma})\left[F(x_{t-1}) - F(x_*)\right] + \gamma\mathcal{O}(k^2 V_1 + V_3 + \epsilon).$$

By telescoping and taking total expectation, after $T$ global updates, we have
$$\mathbb{E}\left[F(x_T) - F(x_*)\right] \leq (1 - \alpha\sqrt{\gamma})^T \left[F(x_0) - F(x_*)\right] + \frac{\sqrt{\gamma}}{\alpha}\mathcal{O}(k^2 V_1 + V_3 + \epsilon).$$

$\square$

For general smooth but non-convex functions, we have the following convergence guarantee.

**Theorem 2.** *Assume that $F(x)$ and $f_s(x)$ are $L$-smooth and potentially non-convex. Assume that for $\forall x$, the true gradients and stochastic gradients are upper-bounded: $\|\nabla F(x)\|^2 \leq V_1$, $\|\nabla f_s(x)\|^2 \leq V_1$, and the stochastic gradients for Zeno testing are always non-zero and lower-bounded: $\|\nabla f_s(x)\|^2 \geq V_2$, where $0 < V_2 \leq V_1$. Furthermore, we assume that the validation set is close to the training set, which implies bounded variance: $\mathbb{E}\left[\|\nabla f_s(x) - \nabla F(x)\|^2\right] \leq V_3, \forall x$. Taking $\gamma < \min(1, \frac{1}{L})$ and $\rho \geq \frac{\alpha\sqrt{\gamma}V_1}{V_2}$, after $T$ global updates, Algorithm 2 converges to a critical point:*

$$\frac{\mathbb{E}\left[\sum_{t\in[T]}\|\nabla F(x_{t-1})\|^2\right]}{T} \leq \frac{\mathbb{E}\left[F(x_0) - F(x_*)\right]}{\alpha\sqrt{\gamma}T} + \frac{\sqrt{\gamma}}{\alpha}\mathcal{O}(k^2V_1 + V_3 + \epsilon).$$

*Furthermore, if we take $\gamma = \frac{1}{LT}$, then we have*

$$\frac{\mathbb{E}\left[\sum_{t\in[T]}\|\nabla F(x_{t-1})\|^2\right]}{T} \leq \mathcal{O}\left(\frac{1}{\alpha\sqrt{T}}\right).$$

*Proof.* Similar to Theorem 1, we have

$$\langle \nabla F(x_{t-1}), -\gamma g \rangle \leq -\rho\frac{V_2}{V_1}\|\nabla F(x_{t-1})\|^2 + \gamma\mathcal{O}(k^2V_1 + V_3 + \epsilon).$$

Using smoothness, taking $\rho \geq \frac{\alpha\sqrt{\gamma}V_1}{V_2}$, we have

$$\mathbb{E}\left[F(x_t)\right]$$

$$\leq F(x_{t-1}) + \langle \nabla F(x_{t-1}), -\gamma\mathbb{E}[g] \rangle + \frac{L\gamma^2}{2}\mathbb{E}\|g\|^2$$

$$\leq F(x_{t-1}) - \rho\frac{V_2}{V_1}\|\nabla F(x_{t-1})\|^2 + \gamma\mathcal{O}(k^2V_1 + V_3 + \epsilon)$$

$$\leq F(x_{t-1}) - \alpha\sqrt{\gamma}\|\nabla F(x_{t-1})\|^2 + \gamma\mathcal{O}(k^2V_1 + V_3 + \epsilon).$$

Thus, we have

$$\|\nabla F(x_{t-1})\|^2 \leq \frac{\mathbb{E}\left[F(x_{t-1}) - F(x_t)\right]}{\alpha\sqrt{\gamma}} + \frac{\sqrt{\gamma}}{\alpha}\mathcal{O}(k^2V_1 + V_3 + \epsilon).$$

By telescoping and taking total expectation, after $T$ global updates, we have

$$\frac{\mathbb{E}\left[\sum_{t\in[T]}\|\nabla F(x_{t-1})\|^2\right]}{T} \leq \frac{\mathbb{E}\left[F(x_0) - F(x_*)\right]}{\alpha\sqrt{\gamma}T} + \frac{\sqrt{\gamma}}{\alpha}\mathcal{O}(k^2V_1 + V_3 + \epsilon).$$

Furthermore, if we take $\gamma = \frac{1}{LT}$, then we have

$$\frac{\mathbb{E}\left[\sum_{t\in[T]}\|\nabla F(x_{t-1})\|^2\right]}{T} \leq \mathcal{O}\left(\frac{1}{\alpha\sqrt{T}}\right).$$

□

## A.2 ZENO+

**Theorem 3.** *Assume that $F(x)$ and $f_s(x)$ have $L$-smoothness has $\mu$-strong convexity. Assume that for $\forall x$, the true gradients and stochastic gradients are upper-bounded: $\|\nabla F(x)\|^2 \leq V_1$, $\|\nabla f_s(x)\|^2 \leq V_1$, and the stochastic gradients for Zeno testing are always non-zero and lower-bounded: $\|\nabla f_s(x)\|^2 \geq V_2$, where $0 < V_2 \leq V_1$. Furthermore, we assume that the validation set is close to the training set, which implies bounded variance: $\mathbb{E}\left[\|\nabla f_s(x) - \nabla F(x)\|^2\right] \leq V_3, \forall x$. Taking $\gamma < \min(1, \frac{1}{L})$ and $\rho \geq \frac{\alpha\sqrt{\gamma}V_1}{2\mu V_2} - \frac{\mu\gamma^2}{2}$, after $T$ global updates, Algorithm 1 converges to a global optimum:*

$$\mathbb{E}\left[F(x_T) - f(x_*)\right] \leq (1 - \alpha\sqrt{\gamma})^T\left[F(x_0) - f(x_*)\right] + \frac{\sqrt{\gamma}}{\alpha}\mathcal{O}(V_1 + V_3 + \epsilon).$$

*Proof.* Using $\mu$-strong convexity, we have

$$\langle \nabla f_s(x), -\gamma g \rangle + \frac{\mu \gamma^2}{2} \|g\|^2 \leq f_s(x - \gamma g) - f_s(x) \leq -\rho \|g\|^2 + \gamma \epsilon.$$

Note that $\frac{\|\nabla f_s(x)\|^2}{\|\nabla F(x)\|^2} \geq \frac{V_2}{V_1}$. Thus, we have

$$\langle \nabla f_s(x), -\gamma g \rangle \leq -\left(\rho + \frac{\mu \gamma^2}{2}\right) \|\nabla f_s(x)\|^2 + \gamma \epsilon \leq -\left(\rho + \frac{\mu \gamma^2}{2}\right) \frac{V_2}{V_1} \|\nabla F(x)\|^2 + \gamma \epsilon.$$

Then, we have

$$\begin{aligned}
&\langle \nabla F(x), -\gamma g \rangle \\
&= \langle \nabla f_s(x), -\gamma g \rangle + \langle \nabla F(x) - \nabla f_s(x), -\gamma g \rangle \\
&= \langle \nabla f_s(x), -\gamma g \rangle + \gamma \langle (\nabla F(x) - \nabla f_s(x)), -g \rangle \\
&\leq \langle \nabla f_s(x), -\gamma g \rangle + \frac{\gamma}{2} \|\nabla F(x) - \nabla f_s(x)\|^2 + \frac{\gamma}{2} \|g\|^2.
\end{aligned}$$

Taking the expectation on both sides, we have

$$\langle \nabla F(x), -\gamma \mathbb{E}[g] \rangle \leq -\left(\rho + \frac{\mu \gamma^2}{2}\right) \frac{V_2}{V_1} \|\nabla F(x)\|^2 + \frac{\gamma}{2}(V_1 + V_3) + \gamma \epsilon.$$

Using $L$-smoothness, conditional on $x$, we have

$$\begin{aligned}
&\mathbb{E}\left[F(x - \gamma g) - F(x)\right] \\
&\leq \langle \nabla F(x), -\gamma \mathbb{E}[g] \rangle + \frac{L \gamma^2}{2} \mathbb{E}\|g\|^2 \\
&\leq -\left(\rho + \frac{\mu \gamma^2}{2}\right) \frac{V_2}{V_1} \|\nabla F(x)\|^2 + \frac{\gamma}{2}(V_1 + V_3) + \frac{L \gamma^2}{2} V_1 + \gamma \epsilon \\
&\leq -\left(\rho + \frac{\mu \gamma^2}{2}\right) \frac{V_2}{V_1} \|\nabla F(x)\|^2 + \gamma \mathcal{O}(V_1 + V_3 + \epsilon).
\end{aligned}$$

Again, using $\mu$-strong convexity, we have

$$F(x) - F(x_*) \leq \frac{1}{2\mu} \|\nabla F(x)\|^2.$$

Thus, we have

$$\mathbb{E}\left[F(x - \gamma g) - f(x_*)\right] \leq \left[1 - 2\mu \left(\rho + \frac{\mu \gamma^2}{2}\right) \frac{V_2}{V_1}\right] [F(x) - f(x_*)] + \gamma \mathcal{O}(V_1 + V_3 + \epsilon).$$

Taking $x_{t-1} = x$, and $x_t = x - \gamma g$, we have

$$\mathbb{E}\left[F(x_t) - f(x_*)\right] \leq \left[1 - 2\mu \left(\rho + \frac{\mu \gamma^2}{2}\right) \frac{V_2}{V_1}\right] [F(x_{t-1}) - f(x_*)] + \gamma \mathcal{O}(V_1 + V_3) + \epsilon.$$

Take $\rho \geq \frac{\alpha \sqrt{\gamma} V_1}{2\mu V_2} - \frac{\mu \gamma^2}{2}$, we have

$$\mathbb{E}\left[F(x_t) - f(x_*)\right] \leq (1 - \alpha \sqrt{\gamma}) [F(x_{t-1}) - f(x_*)] + \gamma \mathcal{O}(V_1 + V_3 + \epsilon).$$

By telescoping and taking total expectation, after $T$ global updates, we have

$$\begin{aligned}
&\mathbb{E}\left[F(x_T) - f(x_*)\right] \\
&\leq (1 - \alpha \sqrt{\gamma})^T [F(x_0) - f(x_*)] + \frac{1 - (1 - \alpha \sqrt{\gamma})^T}{1 - (1 - \alpha \sqrt{\gamma})} \gamma \mathcal{O}(V_1 + V_3 + \epsilon) \\
&\leq (1 - \alpha \sqrt{\gamma})^T [F(x_0) - f(x_*)] + \frac{\sqrt{\gamma}}{\alpha} \mathcal{O}(V_1 + V_3 + \epsilon).
\end{aligned}$$

$\square$

# B  ADDITIONAL EXPERIMENTS

## B.1  NO ATTACK

We first test the convergence when there are no attacks. In all the experiments, we take the learning rate $\gamma = 0.1$, mini-batch size $n = n_s = 128$, $\rho = 0.002$, $\epsilon = 0.1$, $k = 10$. For Kardam, we take

$q = 2$ (Kardam pretends that there are 2 Byzantine workers). The result is shown in Figure 3. We can see that Zeno++ converges a little bit slower than AsyncSGD, but faster than Kardam, especially when the worker delay is large. When $k_w = 10$, Zeno++ converges much faster than Kardam.

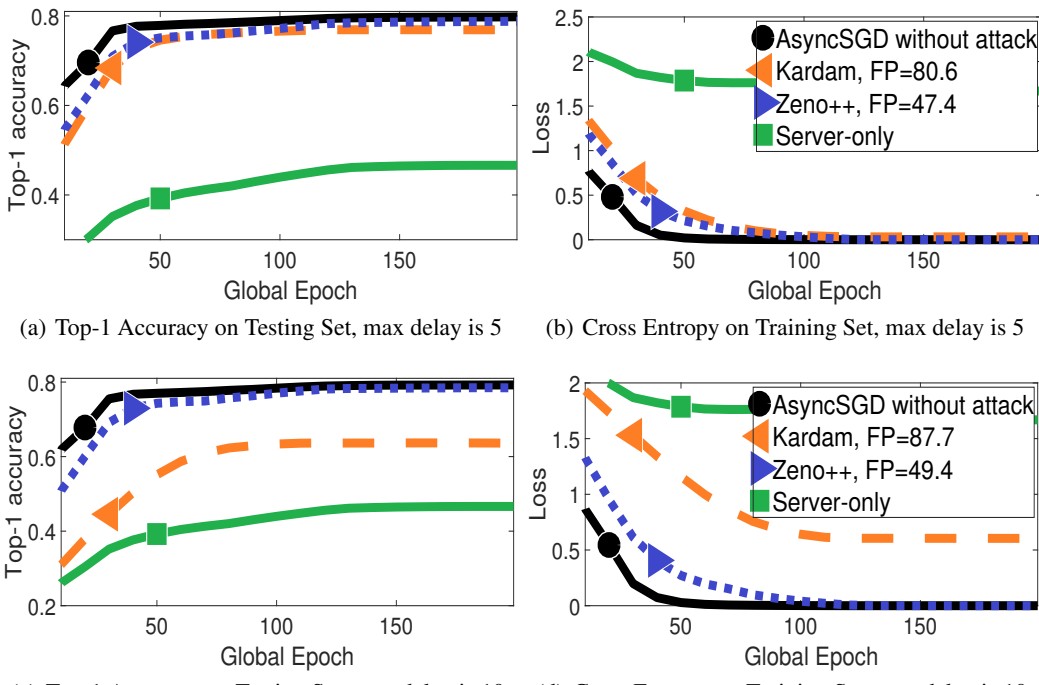

(a) Top-1 Accuracy on Testing Set, max delay is 5    (b) Cross Entropy on Training Set, max delay is 5

(c) Top-1 Accuracy on Testing Set, max delay is 10    (d) Cross Entropy on Training Set, max delay is 10

Figure 3: Convergence without attacks, with different maximum delays of workers. $\rho = 0.002, \epsilon = 0.1$ for Zeno++.

### B.2 SIGN-FLIPPING ATTACK

We test the Byzantine-tolerance to "sign-flipping" attack, which is proposed in Damaskinos et al. (2018). In such attacks, the Byzantine workers send $-10\nabla f(x)$ instead of the correct gradient $\nabla f(x)$ to the server. In all the experiments, we take the learning rate $\gamma = 0.1$, mini-batch size $n = n_s = 128$, $\rho = 0.002$, $\epsilon = 0.1$, $k = 10$. The result is shown in Figure 4 and 5, with different number of Byzantine workers $q$.

### B.3 LABEL-FLIPPING ATTACK

We test the Byzantine tolerance to the label-flipping attacks. When such kind of attacks happen, the workers compute the gradients based on the training data with "flipped" labels, i.e., any $label \in \{0, \ldots, 9\}$, is replaced by $9 - label$. Such kind of attacks can be caused by data poisoning or software failures. In all the experiments, we take the learning rate $\gamma = 0.1$, mini-batch size $n = n_s = 128$, $\rho = 0.002$, $\epsilon = 0.1$, $k = 10$. The result is shown in Figure 6 and 7, with different number of Byzantine workers $q$.

### B.4 SENSITIVITY TO HYPERPARAMETERS

In Figure 8 and 9, we show how the hyperparameters $\rho$, $\epsilon$, and $k$ affect the convergence. In general, Zeno++ is insensitive to $\epsilon$. Larger $\rho$ and $k$ slow down the convergence.

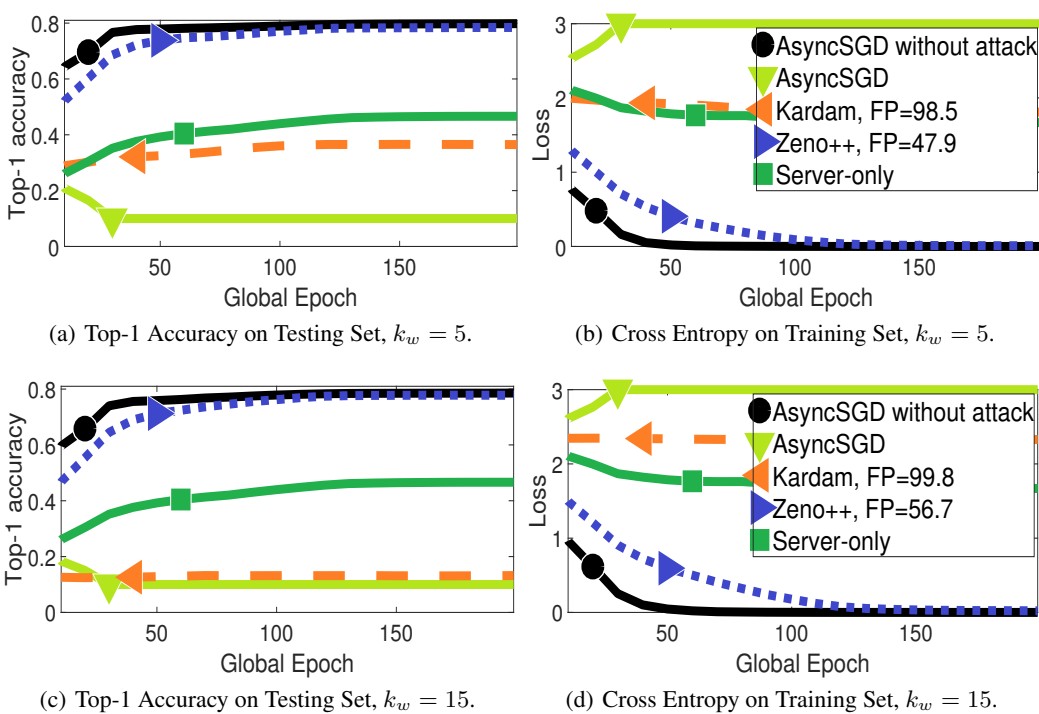

(a) Top-1 Accuracy on Testing Set, $k_w = 5$.

(b) Cross Entropy on Training Set, $k_w = 5$.

(c) Top-1 Accuracy on Testing Set, $k_w = 15$.

(d) Cross Entropy on Training Set, $k_w = 15$.

Figure 4: Convergence with sign-flipping attacks, with different maximum worker delays $k_w$. For any correct gradient $g$, if selected to be Byzantine, $g$ will be replaced by $-10g$. $q = 4$ out of the 10 workers are Byzantine. $\rho = 0.002, \epsilon = 0.1, k = 10$ for Zeno++.

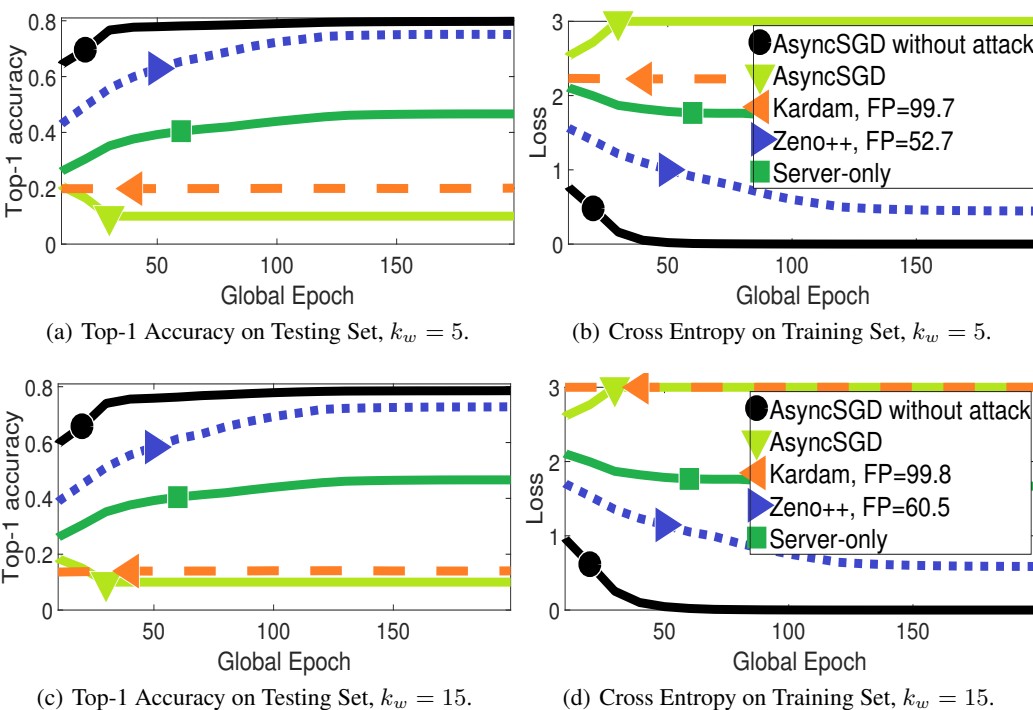

(a) Top-1 Accuracy on Testing Set, $k_w = 5$.

(b) Cross Entropy on Training Set, $k_w = 5$.

(c) Top-1 Accuracy on Testing Set, $k_w = 15$.

(d) Cross Entropy on Training Set, $k_w = 15$.

Figure 5: Convergence with sign-flipping attacks, with different maximum worker delays $k_w$. For any correct gradient $g$, if selected to be Byzantine, $g$ will be replaced by $-10g$. $q = 8$ out of the 10 workers are Byzantine. $\rho = 0.002, \epsilon = 0.1, k = 10$ for Zeno++.

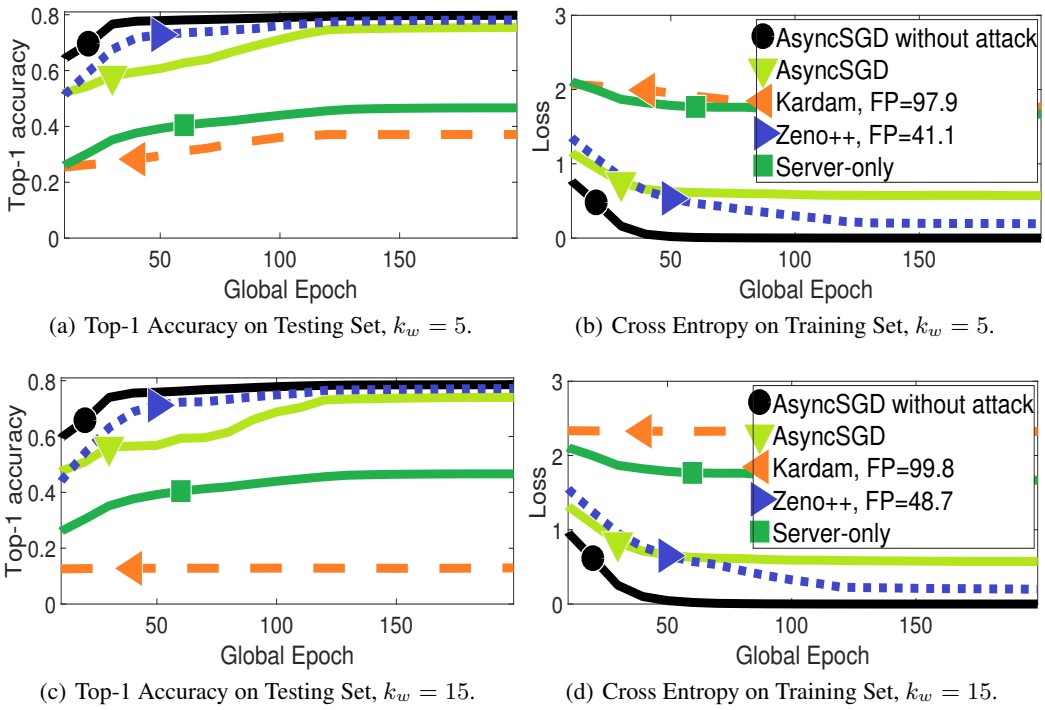

Figure 6: Convergence with label-flipping attacks, with different maximum worker delays $k_w$. $q = 4$ out of the 10 workers are Byzantine. $\rho = 0.002, \epsilon = 0.1, k = 10$ for Zeno++.

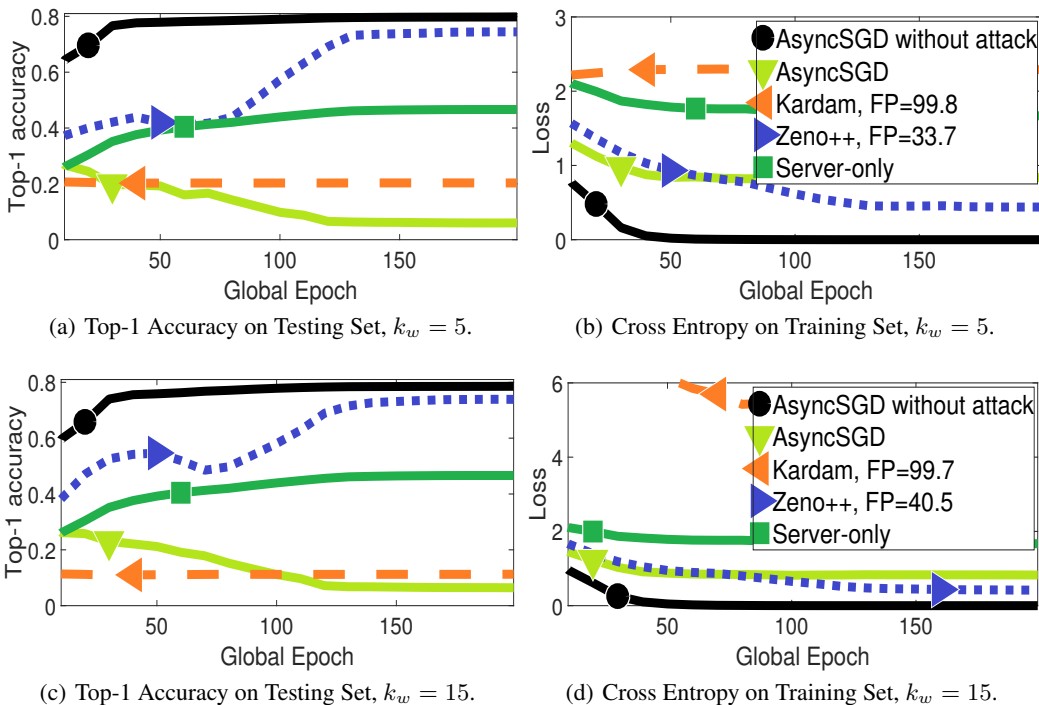

Figure 7: Convergence with label-flipping attacks, with different maximum worker delays $k_w$. $q = 6$ out of the 10 workers are Byzantine. $\rho = 0.002, \epsilon = 0.1, k = 10$ for Zeno++.

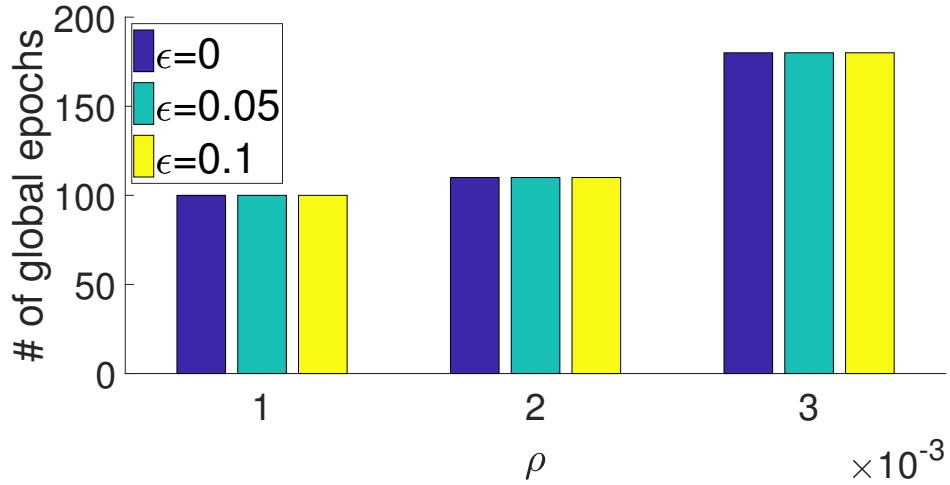

Figure 8: Number of global epochs to reach training loss value $0.2$, with sign-flipping attacks and $q = 4$ Byzantine workers. $k = 10$ for `Zeno++`. $\rho$ and $\epsilon$ varies.

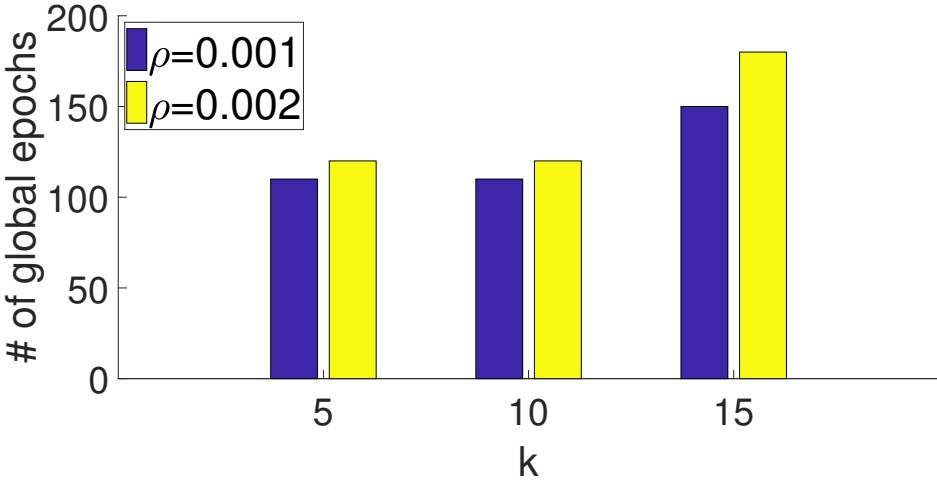

Figure 9: Number of global epochs to reach training loss value $0.2$, with sign-flipping attacks and $q = 6$ Byzantine workers. $\epsilon = 0$ for `Zeno++`. $\rho$ and $k$ varies.

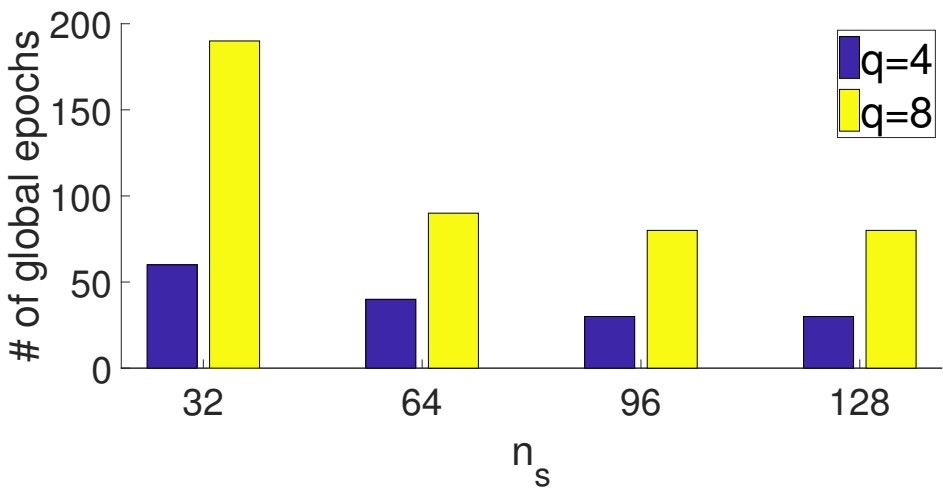

Figure 10: Number of global epochs to reach training loss value $0.7$, with sign-flipping attacks and $q \in \{4, 8\}$ Byzantine workers. $\rho = 0.001$, $\epsilon = 0$, $k = 15$ for Zeno++. The batch size of Zeno++, $n_s$, varies.

### B.5 RESNET20 ON CIFAR-10

We test the Byzantine-tolerance on ResNet20 v1 (https://keras.io/examples/cifar10_resnet/) and CIFAR-10 dataset. We show the top-1 accuracy on the testing set.

In Figure 11, we show the convergence when there are no attacks. `Zeno++` converges as good as `AsyncSGD`. In Figure 12, we show the convergence when there are sign-flipping attacks.

In Figure 13 and 14, we test `Zeno++` with 2 different types of random attacks. In Figure 13, for any correct gradient $g$, if selected to be Byzantine, any element of $g$ will be **replaced** by a IID random value drawn from a Gaussian distribution with 0 mean and 5 variance, while in Figure 14, any element of Byzantine $g$ will be **added** by a IID random value drawn from a Gaussian distribution with 0 mean and 5 variance.

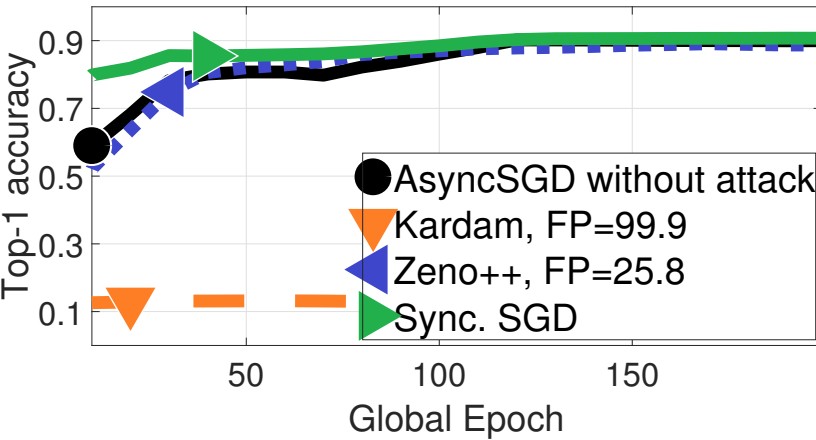

Figure 11: Convergence (top-1 accuracy on the testing set) without attacks, with 20 workers and maximum worker delays $k_w = 15$. $\rho = 0.00005, \epsilon = 0, k = 15$ for `Zeno++`.

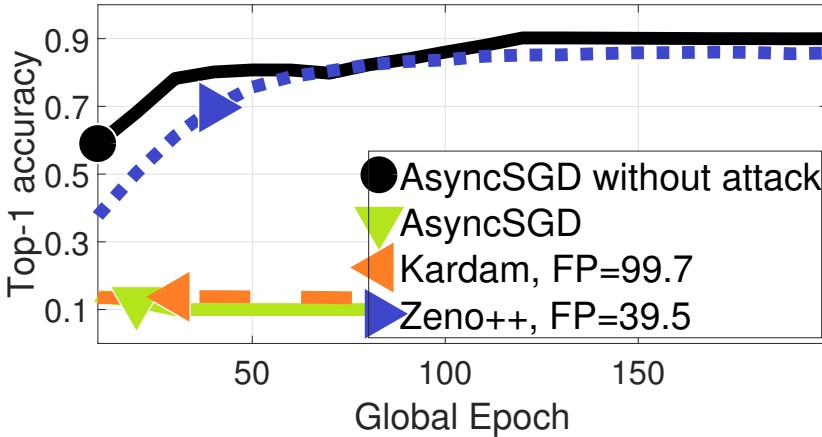

Figure 12: Convergence (top-1 accuracy on the testing set) with sign-flipping attacks, with 20 workers and maximum worker delays $k_w = 15$. For any correct gradient $g$, if selected to be Byzantine, $g$ will be replaced by $-10g$. $q = 12$ out of the 20 workers are Byzantine. $\rho = 0.00005, \epsilon = 0, k = 15$ for `Zeno++`.

### B.6 LSTM ON WIKITEXT-2

We test the Byzantine-tolerance on LSTM-based language model. The model is from https://github.com/salesforce/awd-lstm-lm (Merity et al., 2017; 2018), trained on WikiText-2 dataset. We use 600 hidden units per layer. We show the perplexity (the smaller the better) on the testing set.

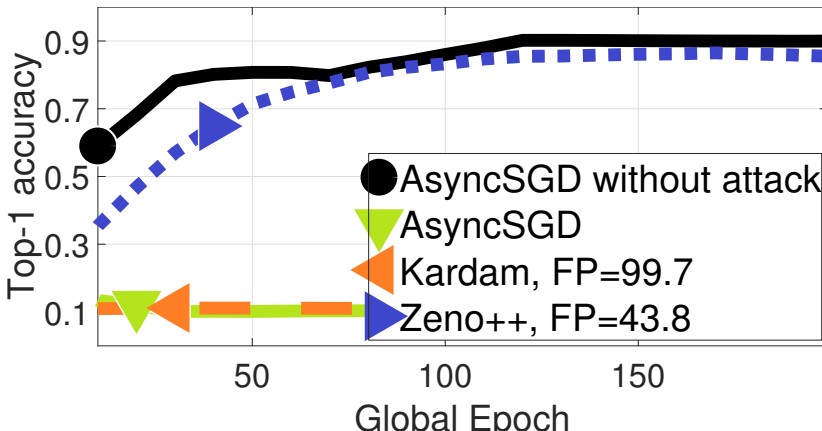

Figure 13: Convergence (top-1 accuracy on the testing set) with random attacks (type I), with 20 workers and maximum worker delays $k_w = 15$. For any correct gradient $g$, if selected to be Byzantine, any element of $g$ will be replaced by a IID random value drawn from a Gaussian distribution with 0 mean and 5 variance. $q = 12$ out of the 20 workers are Byzantine. $\rho = 0.00005, \epsilon = 0, k = 15$ for Zeno++.

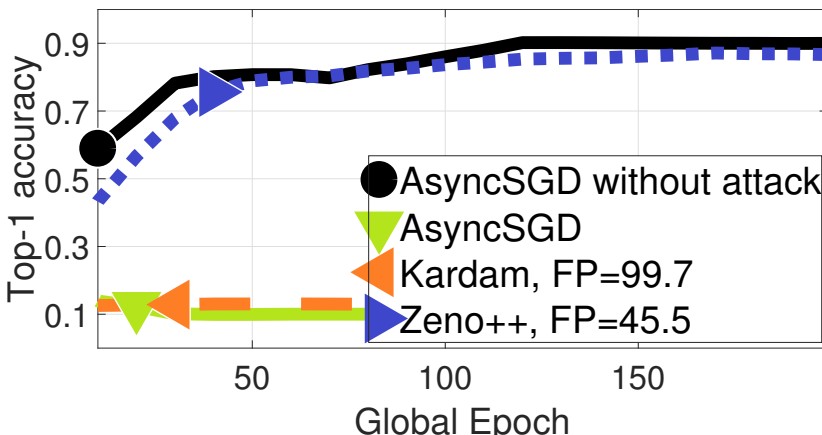

Figure 14: Convergence (top-1 accuracy on the testing set) with random attacks (type II), with 20 workers and maximum worker delays $k_w = 15$. For any correct gradient $g$, if selected to be Byzantine, any element of $g$ will be added by a IID random value drawn from a Gaussian distribution with 0 mean and 5 variance. $q = 12$ out of the 20 workers are Byzantine. $\rho = 0.00005, \epsilon = 0, k = 15$ for Zeno++.

In Figure 15, we show the convergence when there are no attacks. Zeno++ converges as good as AsyncSGD. In Figure 16 and 17, we show the convergence when there are sign-flipping attacks, with different number of Byzantine workers.

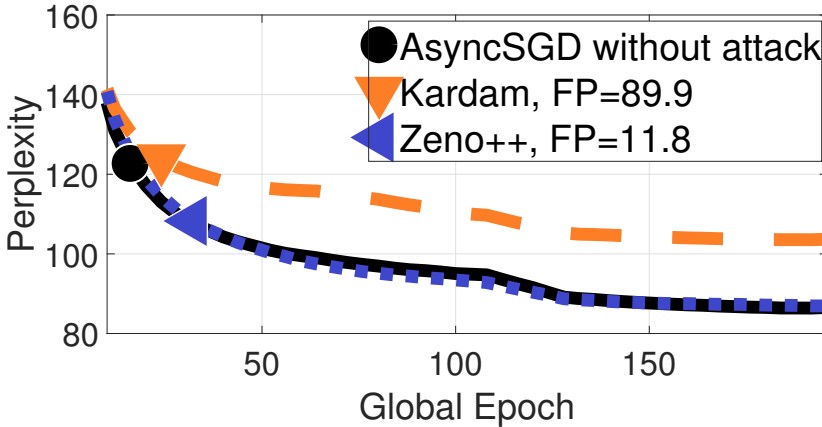

Figure 15: Convergence (perplexity on the testing set) without attacks, with 10 workers and maximum worker delays $k_w = 5$. $\rho = 0.005, \epsilon = 0.001, k = 10$ for `Zeno++`.

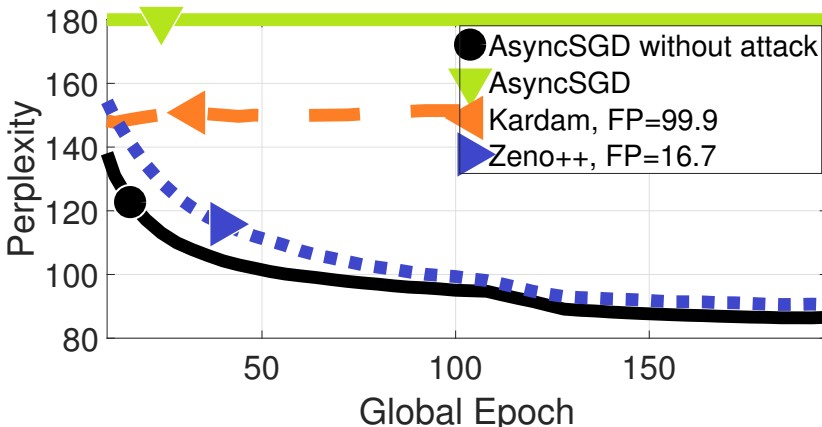

Figure 16: Convergence (perplexity on the testing set) with sign-flipping attacks, with 10 workers and maximum worker delays $k_w = 5$. For any correct gradient $g$, if selected to be Byzantine, $g$ will be replaced by $-10g$. $q = 4$ out of the 10 workers are Byzantine. $\rho = 0.005, \epsilon = 0.001, k = 10$ for `Zeno++`.

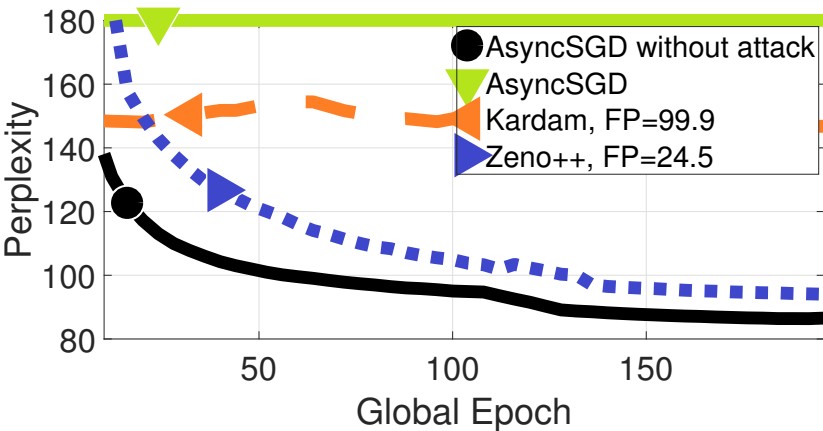

Figure 17: Convergence (perplexity on the testing set) with sign-flipping attacks, with 10 workers and maximum worker delays $k_w = 5$. For any correct gradient $g$, if selected to be Byzantine, $g$ will be replaced by $-10g$. $q = 6$ out of the 10 workers are Byzantine. $\rho = 0.005, \epsilon = 0.0004, k = 10$ for `Zeno++`.

