# OpenReview forum: "Zeno++: Robust Fully Asynchronous SGD"
_ICLR.cc/2020/Conference — Reject_

### Official Review · AnonReviewer3 · 2019-10-22
**Official Blind Review #3**

**Rating:** 3

**Review:**

This paper addresses security of distributed optimization algorithm under Byzantine failures. These failures usually prevent convergence of training neural network models. Focusing on the asynchronous SGD algorithm implemented with a parameter-server, the authors propose to use stochastic line search ideas to detect whether the gradients are good descent directions or not. It applies to a general scenario including repeated and unbounded Byzantine failures.

Theoretical results of this paper seem to me to have some flaw. In the proof of Theorem 1, line 6, it is not clear to me why the gradient norm ||g||^2 is replaced by || grad f_s (x_tau) ||^2. Clearly the g comes from any worker which can be very different to grad f_s (x_tau). Therefore, I recommend the authors check the proof of both theorem 1 and 2 more carefully.

Numerical results show that the proposed algorithm Zeno++ works well with sign-flipping attacks. However, this scenario is very limited to validate all imaginable Byzantine failures that this paper would like to address. For example, one can use random gradients instead of sign-flipped gradients as a Byzantine attack, would the algorithm still work?


**Experience Assessment:**

I have published one or two papers in this area.

**Review Assessment: Checking Correctness Of Derivations And Theory:**

I carefully checked the derivations and theory.

**Review Assessment: Checking Correctness Of Experiments:**

I assessed the sensibility of the experiments.

**Review Assessment: Thoroughness In Paper Reading:**

I read the paper at least twice and used my best judgement in assessing the paper.

---

> ### Author Response · Authors · 2019-11-08
> **Authors' feedback**
>
> 1.	Question:“In the proof of Theorem 1, line 6, it is not clear to me why the gradient norm ||g||^2 is replaced by || grad f_s (x_tau) ||^2.”
> Answer: This is because of (re-)normalization. In Zeno++, Line 4-6 of Algorithm 2 (Server), the server receives $\tilde{g}$ from worker, and then normalize it, which results in $g$, where $\|g\|^2 = \| \nabla f_s(x_\tau) \|^2$. It is true that $g$ is very different from $\nabla f_s(x_\tau) $, but our algorithm makes them to have the same 2-norm by normalization.
> The purpose/intuition of doing so is that the attackers can also rescale the candidate gradient $\tilde{g}$ to have very large 2-norm, which will be very harmful to the convergence. By normalization, $g$ will have similar 2-norm as ordinary gradients, so that even if a Byzantine $g$ passes the test of Zeno+ or Zeno++, the harm will be limited.
>
> We have added some experiments according to the reviewer's requirement, including experiments with random attack.
> We have also double-checked the proof, and we think the current version is correct.
>
> We hope that our answers and clarifications resolve the reviewers' concern.

---

> ### Author Response · Authors · 2019-11-11
> **Added experiments with random attacks**
>
> Yes, our proposed algorithm is supposed to tolerate any type of attacks, since Byzantine attacks are supposed to be arbitrary.
> According to the reviewer's requirement, we add the following experiments with random attacks (appended to the end of appendix, in Section B.5, Figure 13 and Figure 14):
> The experiments are conducted on ResNet20 v1 (from https://keras.io/examples/cifar10_resnet/) with CIFAR-10 dataset.
>
> We test Byzantine tolerance on 2 different types of random attacks.
> Type I: For any correct gradient $g$, if selected to be Byzantine, $g$ will be replaced by IID random values drawn from a Gaussian distribution with 0 mean and 5 variance.
> Type II: For any correct gradient $g$, if selected to be Byzantine, $g$ will be replaced by $g + \delta$, where $\delta$ is a IID random vector drawn from a Gaussian distribution with 0 mean and 5 variance.
>
>
> Please check the revised manuscript for the new empirical results.

---

### Official Review · AnonReviewer2 · 2019-10-23
**Official Blind Review #2**

**Rating:** 3

**Review:**

Summary:
This paper investigates the security of distributed asynchronous SGD. Authors propose Zeno++, worker-server asynchronous implementation of SGD which is robust to Byzantine failures. To ensure that the gradients sent by the workers are correct, Zeno++ server scores each worker gradients using a “reference” gradient computed on a “secret” validation set.  If the score is under a given threshold, then the worker gradient is discarded.

Authors provide convergence guarantee for the Zeno++ optimizer for non-convex function. In addition, they provide an empirical evaluation of Zeno++ on the CIFAR10 datasets and compare with various baselines.

Originality:
I would argue that the paper novelty is limited.  Paper builds upon the Zeno algorithm.  From the paper, the main changes with respect to Zeno algorithm is the use of a hard-threshold instead of a ranking to adapt Zeno to the asynchronous case, and the use of a first-order Tayler approximation of the score to speed up its computations.

Clarity:
Paper is clear and easy to follow.

Quality:
My main concerns are related to the experimental section. Authors only report results for one model on one dataset. It is unclear how those results would transfer to different tasks and architectures. In addition, the experiments are relatively small scale (10 workers), how does the system scale as you increase the number of workers?

The top-1 reported on CIFAR10 seems pretty low with respect to the current state-of-art. It would be nice to use a more competitive model such as a Resnet-18 to verify that one can achieve similar performance with Zeno++ compared to AR-SGD (without attack).

Authors introduced Zeno++ to reduce the computation overhead over Zeno+. Did you empirically quantify this speed-up, and do you achieve similar accuracy than Zeno+?

Significance:
Authors only compare with asynchronous baseline. It would be nice to demonstrate the advantage of asynchronous methods over synchronous one (such as Zeno and All-Reduce SGD without attack). Can you show a speed benefit of asynchronous Zeno++ and show similar accuracy than synchronous approaches?

Minor:
-	It would be to reports use a log scale in Fig 1 b), c) and Fig 2. b), d).


**Experience Assessment:**

I do not know much about this area.

**Review Assessment: Checking Correctness Of Derivations And Theory:**

I did not assess the derivations or theory.

**Review Assessment: Checking Correctness Of Experiments:**

I carefully checked the experiments.

**Review Assessment: Thoroughness In Paper Reading:**

I read the paper thoroughly.

---

> ### Author Response · Authors · 2019-11-08
> **Authors' feedback**
>
> 1.	For novelty, we have to argue that asynchronous training is very different from synchronous training. Although the main idea is based on Zeno, applying such idea to asynchronous training requires a lot of non-trivial practical and theoretical work, which is our contribution. Furthermore, as we have shown in the experiments, since the performance of the baseline (Kardam) is very bad, our proposed algorithm could be the first algorithm that actually tolerates Byzantine failures and makes reasonable progress in convergence for asynchronous training.
>
> 2.	Asynchronous training and synchronous training are designed for different scenarios. When there are stragglers in the workers, asynchronous training is preferred. In this paper, we only answer the question “if we use asynchronous training, can we develop an algorithm to tolerate Byzantine workers?”, and assume that there are stragglers in the workers, so that asynchronous training is preferred.
> Even if all the workers are assumed to be homogeneous, “which one of synchronous training and asynchronous training is faster” is a controversial topic in distributed machine learning. There are tricks (e.g., rescaling learning rate w.r.t. batch sizes, and warmup) to improve the performance of synchronous SGD to match the performance with single-threaded SGD with small batch sizes. There are also tricks [2] to improve asynchronous SGD to match the performance of synchronous SGD. We believe that the research and discussion about the competition between synchronous training and asynchronous training will continue for a long time, which is out of the scope of this paper.
>
> 3.	It is perhaps worth emphasizing that Zeno+ is not a previously existing algorithm i.e. not a previous baseline. Zeno++ is our main contribution (which we evaluate theoretically and empirically). As stated, this novel approach is inspired by Zeno+, which we chose to include for completeness.
> We did not empirically compare Zeno+ and Zeno++, but we can theoretically compare the computation overhead on the server side.
> Assume that model size is $d$. Assume that the overhead of simple element-wise vector operations is $c_1 d$, the overhead of a single forward step is $c_2 d$, the overhead of a single backward step is $c_3 d$. When using Zeno+ or Zeno++, the server uses $n_s$ samples to validate any received gradient candidate. Zeno++ updates the validation vector $v$ after receiving every $k$ gradient candidates. Typically, $c_1 \leq c_2 \leq c_3$
> For Zeno+, the overhead of validation is $2 c_1 d + n_s c_2 d$ ($2 c_1$ for computing $x-\gamma g$ and $\|g\|^2$, evaluating $f_S(x-\gamma g)$ takes $n_s$ forward steps, ignoring the overhead of computing $f_s(x)$). Furthermore, note that the validation can only be started after the gradient candidate $g$ is received.
> For Zeno++, the overhead of validation on average is $2 c_1 d + n_s (c_2 + c_3) d / k$ ($2 c_1$ for computing $<v, g>$ and $\|g\|^2$, computing $v$ takes $n_s$ backward and forward steps for every $k$ iterations). Furthermore, note that the computation of $v$ does not need to wait until $g$ is received.
> Thus, if we assume $c_2 \approx c_3$, then taking $k>2$ will make the overhead of Zeno++ less than that of Zeno+. Note that in our experiments, we take $k=10$. Furthermore, since for Zeno++, the computation of $v$ is non-blocking, the overhead $n_s (c_2 + c_3) d / k$ could be hidden. In the ideal cases, the overhead of Zeno++ could be very close to $2 c_1 d$.
>
> 4. We are adding some experiments according to the reviewer's requirement, including experiments on Resnet-20, and experiments on language models. We will revise the manuscript and append the new results after the additional experiments are done.
>
> References
> [1] You, Yang, et al. "Imagenet training in minutes." Proceedings of the 47th International Conference on Parallel Processing. ACM, 2018.
> [2] Aji, Alham Fikri, and Kenneth Heafield. "Making Asynchronous Stochastic Gradient Descent Work for Transformers." arXiv preprint arXiv:1906.03496 (2019).

---

> > ### Comment · AnonReviewer2 · 2019-11-15
> > **Acknowledging responses**
> >
> > Thank you for your rebuttal. I appreciate the additional experiments and the clarification about the computation overhead of Zeno+.
> >
> > However, it appears the accuracy or perplexity reported on B.5 ans B.6 are worse than what is reported in previous works. I understand that the goal of the paper is not to achieve state-of-art performance on those datasets, but to show to develop an asynchronous approach robust to Byzantine failures. However, given that it degrades the performance compared to synchronous-SGD, the significance of Zeno++  remains a bit unclear to me.

---

> > > ### Author Response · Authors · 2019-11-15
> > > **Comparing to synchronous SGD is unfair**
> > >
> > > Thanks for the feedback.
> > > We want to highlight the following comments:
> > >
> > > 1. It is unfair to compare asynchronous SGD with synchronous SGD, since they are used in different scenarios. We use asynchronous SGD when there are extremely slow stragglers in the workers. Synchronous SGD should be used when the workers are homogeneous.
> > >
> > > 2. The significance of Zeno++ is that it tolerates Byzantine attacks theoretically and empirically, and achieves much better performance compared to the baselines when there are Byzantine attacks.
> > >
> > > 3. We want to highlight that the major contribution of this paper is the theoretical guarantees. The empirical results are only used to validate the theoretical results.
> > >
> > > 4. The result in B.5 (accuracy around 90%, for asynchronous SGD without attack) is very close to the state-of-art performance (91%~92%). Note that the state-of-art performance is a result of a long history of fine-tuning the hyperparameters. However, there is very few work reporting how to fine-tune the performance of asynchronous SGD. So, it's very likely that we simply didn't find the best learning rates (although we have already ) and random seeds for the baseline. However, with the same hyperparameters, Zeno++ achieves almost the same performance when there are no attacks, and much better performance when there are Byzantine attacks.
> > >
> > > 5. In B.6, we actually achieve the state-of-art performance. We use a model smaller than the original paper ( https://github.com/salesforce/awd-lstm-lm ). In the original paper, the number of hidden units per layer is 1150, trained for 750 epochs. In our experiments in B.6, the number of hidden units per layer is only 600, trained for only 200 epochs. The reason we did so is that the original model is too large and time-consuming, and we don't have enough time for that.
> > > In our experiment, we achieve ppl 85.5 at the 200th epoch, which is the same as the one in the following log (ppl 85.46): https://github.com/dmlc/web-data/blob/master/gluonnlp/logs/language_model/awd_lstm_lm_600_wikitext-2.log , which is the state-of-art result from https://gluon-nlp.mxnet.io/model_zoo/language_model/index.html .
> > >
> > > Note that the "perplexity on the testing set" in B.6 is actually the performance on the validation set of Wikitext-2, but we have already used the term "validation dataset for Zeno++" in Section 4.2, so we call the validation set of Wikitext-2 as "testing set" to avoid conflict.

---

> > > ### Author Response · Authors · 2019-11-15
> > > **Synchronous SGD also degrades the performance when scaling to larger batch sizes or more workers**
> > >
> > > We add the baseline of synchronous SGD in Section B.5. Unfortunately, we only have time for the experiment without Byzantine attacks.
> > >
> > > For the reviewer's question about the comparison between synchronous SGD and asynchronous SGD, we have the following additional comment:
> > >
> > > 1. As reported in other previous work [3], when scaling to larger mini-batch sizes or more workers, synchronous SGD also degrades the performance a little bit.
> > > In the original paper of Resnet20, the batch size is 128, the learning rate is 0.1, and the testing accuracy is 91.25%.
> > > In our experiment (Figure 11, in Section B.5, without attacks), the actual batch size is $128$ samples $\times$ $10$ workers = $1280$ samples, the learning rate is re-scaled to $lr = 1$, and the testing accuracy is 90.8%. For the AsyncSGD without attack, we use batch size $128$ and learning rate $0.08$, and the accuracy is 90.4%. For Zeno++ without attack, we use the same hyperparameters as AsyncSGD, and the accuracy is 90.1%.
> > >
> > > Thus, when scaling to more workers, both synchronous SGD and asynchronous SGD degrades the performance. Synchronous SGD degrades because of the larger batch sizes. Asynchronous SGD degrades because of the asynchrony. And our experiment shows that the gap between their performance is tiny.
> > >
> > > 2. When there are no attacks, Zeno++ performs slightly worse than AsyncSGD, because of the false-positives in filtering the Byzantine gradients. That is an inevitable trade-off between convergence and Byzantine-tolerance.
> > >
> > > 3. When there are Byzantine attacks, in each epoch, the server receives the same number of gradients as in the case of "AsyncSGD without attack". However, only 40% of the these gradients are non-Byzantine. Thus, with the same number of epochs, Zeno++ inevitably converges slower than "AsyncSGD without attack" or "Synchronous SGD".
> > >
> > > 4. We focus on the Byzantine-tolerance in this paper. The comparison between synchronous SGD and asynchronous SGD is irrelevant to the significance of Zeno++.
> > > Zeno++ should only be compared to asynchronous SGD, and Kardam, which is the state-of-art Byzantine-tolerant asynchronous SGD algorithm. We have already shown much better performance compared to these baselines.
> > >
> > > References
> > > [3] You, Yang, Igor Gitman, and Boris Ginsburg. "Scaling sgd batch size to 32k for imagenet training." arXiv preprint arXiv:1708.03888 6 (2017).

---

> ### Author Response · Authors · 2019-11-11
> **Added experiments on ResNet20**
>
> According to the reviewer's requirement, we add the following experiments (appended to the end of appendix, in Section B.5 and B.6, Figure 11-17):
>
> 1. Experiments with 20 workers
> 2. Experiments on ResNet20 v1 (from https://keras.io/examples/cifar10_resnet/ ) with CIFAR-10 dataset. We can see that when there are no attacks, Zeno++ converges as good as vanilla asynchronous SGD. When there are Byzantine attacks, Zeno++ converges inevitably slower compared to the non-Byzantine cases, but still makes reasonable progress.
> 3. Experiments on LSTM-based language model (from https://github.com/salesforce/awd-lstm-lm ) with WikiText-2 dataset. We use 600 hidden units per layer. We report the perplexity~(the smaller the better) on the testing set. We can see that on the language model, Zeno++ has similar results compared to the experiments on image classification.
>
>
> Please check the revised manuscript for the new empirical results.

---

### Official Review · AnonReviewer1 · 2019-10-23
**Official Blind Review #1**

**Rating:** 6

**Review:**


This paper proposes an approach to Byzantine fault tolerance in asynchronous distributed SGD. The approach appears to be novel. Theoretical convergence guarantees are provided, and an empirical evaluation illustrates very promising results.

Overall, the paper is well-written and the results appear to be novel and interesting. Although I have a few questions, listed below, I generally lean towards accepting this paper.


The assumption of a lower bound on validation gradients is somewhat troubling, especially for over-parameterized problems where so-called "interpolation" may be possible. I realize that validation samples are never used explicitly for stochastic gradient updates, but the algorithm does ensure that the stochastic gradients used are similar to gradients of validation samples. If one is converging to a (local) minimizer, one wants the gradient to vanish. How do we reconcile these points? Also, to properly set $\rho$, for the theory to be valid, one needs to know this bound (or a lower bound on $V_2$). Is this practical?

The paper claims that the computational overhead of Zeno+ is too great to evaluate for comparison with Zeno++. From my reading of the two methods, it isn't immediately obvious to me why this is the case. Including experiments which at least compare the per-iteration runtime (even if not running Zeno+ for training to completion) would make the paper more compelling. After all, Zeno++ still involves periodically evaluating the gradient at a validation sample.

The paper makes the reasonable point that it is not reasonable to assume a bounded number of adversaries in the asynchronous setting, and the theorem statements make no assumption about the number of adversaries or rate at which received gradients are from a Byzantine worker. However, there are also no guarantees about whether the algorithm will ever make progress (i.e., will line 8 ever be reached?). This should be stated more transparently in the paper. Also, I was wondering, given that a gradient has been computed on the parameter server's validation set, which is assumed to be "clean", why not take a step using this gradient when the test in line 7 fails?

Finally, the paper titles includes SGD, but the description in Def 1 doesn't appear to involve stochastic gradients. Typical parameter server implementations have workers compute mini-batch stochastic gradients, not full gradients on their shard of the training set. Does Zeno++ need to be modified to run in this setting? Does the theory still hold?


Minor:
- Is there a typo in line 5 of Zeno++? Should this be $\nabla f_s$ instead of $f_s$? Otherwise, what does it mean to take the inner product of $v$ and $g$ in line 7?


**Experience Assessment:**

I have read many papers in this area.

**Review Assessment: Checking Correctness Of Derivations And Theory:**

I assessed the sensibility of the derivations and theory.

**Review Assessment: Checking Correctness Of Experiments:**

I carefully checked the experiments.

**Review Assessment: Thoroughness In Paper Reading:**

I read the paper at least twice and used my best judgement in assessing the paper.

---

> ### Author Response · Authors · 2019-11-08
> **Authors' feedback**
>
> 1.	“If one is converging to a (local) minimizer, one wants the gradient to vanish. How do we reconcile these points?”
> Note that we assume that the training data (for $F(x)$) and the validation data (for $f_s(x)$) are different, although they could be similar to each other. Thus, a reasonable implication is that $F(x)$ and $E[ f_s(x) ]$ has different minimizers. Thus, when the model converges to the training data, i.e. $\| \nabla F(x_*) \|^2 = 0$ where $x_*$ is the minimizer, we should have $\| E[ \nabla f_s(x_*) ] \|^2 \neq 0$ since $x_*$ is not a minimizer of $E[ f_s(x) ]$.
> Furthermore, even if the expectation $E[ \nabla f_s(x_*) ] = 0$ $\| E[ \nabla f_s(x_*) ] \|^2 = 0$), the stochastic gradient $\nabla f_s(x_*)$ won’t converge to 0, because $ E[ \| \nabla f_s(x_*) \|^2 ] = E[ \| \nabla f_s(x_*) - E[ \nabla f_s(x_*) ] \|^2 ] + \| E[ \nabla f_s(x_*) ] \|^2 =  E[ \| \nabla f_s(x_*) - E[ \nabla f_s(x_*) ] \|^2 ]$ converges to a non-zero variance.
> Thus, such assumption does not conflict with the convergence/vanishing gradient on the training data.
>
> 2.	For computation overhead:
> It is perhaps worth emphasizing that Zeno+ is not a previously existing algorithm i.e. not a previous baseline. Zeno++ is our main contribution (which we evaluate theoretically and empirically). As stated, this novel approach is inspired by Zeno+, which we chose to include for completeness.
> We did not empirically compare Zeno+ and Zeno++, but we can theoretically compare their computation overhead.
> Assume that model size is $d$. Assume that the overhead of simple element-wise vector operations is $c_1 d$, the overhead of a single forward step is $c_2 d$, the overhead of a single backward step is $c_3 d$. When using Zeno+ or Zeno++, the server uses $n_s$ samples to validate any received gradient candidate. Zeno++ updates the validation vector $v$ after receiving every $k$ gradient candidates. Typically, $c_1 \leq c_2 \leq c_3$
> For Zeno+, the overhead of validation is $2 c_1 d + n_s c_2 d$ ($2 c_1$ for computing $x-\gamma g$ and $\|g\|^2$, evaluating $f_S(x-\gamma g)$ takes $n_s$ forward steps, ignoring the overhead of computing $f_s(x)$). Furthermore, note that the validation can only be started after the gradient candidate $g$ is received.
> For Zeno++, the overhead of validation on average is $2 c_1 d + n_s (c_2 + c_3) d / k$ ($2 c_1$ for computing $<v, g>$ and $\|g\|^2$, computing $v$ takes $n_s$ backward and forward steps for every $k$ iterations).Furthermore, note that the computation of $v$ does not need to wait until $g$ is received.
> Thus, if we assume $c_2 \approx c_3$, then taking $k>2$ will make the overhead of Zeno++ less than that of Zeno+. Note that in our experiments, we take $k=10$. Furthermore, since for Zeno++, the computation of $v$ is non-blocking, the overhead $n_s (c_2 + c_3) d / k$ could be hidden. In the idea cases, the overhead of Zeno++ could be very close to $2 c_1 d$.
>
> 3.	As mentioned by the reviewer, it is true that line 8 of Algorithm 2 may never be reached, if bad hyperparamers ($\rho$ and $\epsilon$) are taken. In our experiments, we found that typically, we can take $\epsilon$ close to 0, and $\rho \in [\gamma \times 10^{-2}, \gamma \times 10^{-1}]$, where $\gamma$ is the learning rate.
>
> 4.	“why not take a step using this gradient when the test in line 7 fails?” According to our objective function, we want to train a model on the training data. We assume that the validation dataset for the testing gradient is similar but different from the training data in expectation (or, they have similar but different distributions). It is possible to use these gradients, but that will end up with training a model on a different objective function.  Thus, in general, we do not want to directly use these gradients for training. Furthermore, in our experiments, we show that if we train the model only on the validation data (no training data is used), the performance will be very bad (see the baseline “Server-only” in all the figures).
>
> 5.	In Definition 1, $\nabla f(x_\tau; z_{i,j})$ is a stochastic gradient. $n$ is the mini-batch size of the stochastic gradient. $z_{i,j}$ is a random sample from the local dataset on the $i$th worker. Also, in Algorithm 2, line 4 of Worker, we randomly draw $n$ samples from the local dataset $D_i$, and compute the stochastic gradient $\tilde{g}$. All the gradients in the algorithms are stochastic, no full gradients are used. The theorems are already for stochastic gradients. We will clearly state that they are all stochastic gradients in a revised version.
>
> 6.	Yes, line 5 of Algorithm 2 is a typo, it should be $\nabla f_s(x_\tau)$ instead of $f_s(x_\tau)$.
>
> 7.	Yes, $<x, y>$ means the inner-product between vector $x$ and $y$.

---

> > ### Comment · AnonReviewer1 · 2019-11-13
> > **Acknowledging responses**
> >
> > Thank you for your responses and clarifications.

---

### Decision · Program_Chairs · 2019-12-19

**Decision:**

Reject

**Comment:**

Main content:

Blind review #2 summarizes it well:

This paper investigates the security of distributed asynchronous SGD. Authors propose Zeno++, worker-server asynchronous implementation of SGD which is robust to Byzantine failures. To ensure that the gradients sent by the workers are correct, Zeno++ server scores each worker gradients using a “reference” gradient computed on a “secret” validation set.  If the score is under a given threshold, then the worker gradient is discarded.

Authors provide convergence guarantee for the Zeno++ optimizer for non-convex function. In addition, they provide an empirical evaluation of Zeno++ on the CIFAR10 datasets and compare with various baselines.

--

Discussion:

Reviews are generally weak on the limited novelty of the approach compared with Zeno, but the rebuttal of the authors on Nov 15 is fair (too long to summarize here).

--

Recommendation and justification:

I do not feel strongly enough to override the weak reviews (but if there is room in the program I would support a weak accept).